# GoalZero: Model-based Hierarchical Self-Play for Sequential Stochastic Combinatorial Optimization

## Abstract

Sequential Stochastic Combinatorial Optimization (SSCO) problems are challenging for reinforcement learning due to exponentially large action spaces, stochastic dynamics, and the need for long-horizon planning under limited resources. Hierarchical Reinforcement Learning (HRL) offers a natural decomposition, but the high-level policy operates in a Semi-Markov Decision Process (SMDP) where actions have variable durations. This variability complicates learning a planning-ready world model. We introduce *GoalZero*, a model-based HRL framework that directly addresses this challenge. GoalZero integrates a MuZero-style planner at the high level that learns a world model of SMDP dynamics. At the core is a principled framework for *multi-timescale SMDP* (MTS-SMDP) world-model learning. Through complementary objectives, the agent learns dynamics where the *latent transition magnitude* correlates with the temporal scale of the corresponding subgoal, facilitating planning over diverse, adaptive temporal abstractions in our evaluated settings. In addition, we propose a subgoal-conditioned budget allocation mechanism learned jointly with the multi-timescale world model, facilitating context-aware resource management. We demonstrate that GoalZero outperforms strong baselines on challenging SSCO benchmarks.

## 1 Introduction

Sequential Stochastic Combinatorial Optimization (SSCO) problems involve making a sequence of combinatorial decisions under uncertainty to maximize a cumulative objective (Li et al., 2018; Kool et al., 2019). These problems underlie applications such as dynamic vehicle routing (Kool et al., 2019) and adaptive influence maximization in social networks (Chen et al., 2021). SSCO is especially challenging because the action space is combinatorial, the dynamics are stochastic and path-dependent, and early decisions can have long-lasting effects. For example, in **Adaptive Influence Maximization (AIM)**, an agent repeatedly chooses seed nodes to trigger stochastic cascades; in the **Stochastic Orienteering Problem (SOP)**, an agent plans multi-day routes on a graph under tight travel budgets. In both cases, strategic aims such as "target a specific community" or "sweep a profitable region" may require very different amounts of effort to carry out.

Flat reinforcement learning (RL) methods often struggle in these settings: they must act directly in an enormous action space and solve long-horizon credit assignment without explicit structure. Hierarchical RL (HRL) (Dayan & Hinton, 1992; Barto & Mahadevan, 2003; Sutton et al., 1999) offers a natural way to introduce temporal abstraction, but in SSCO it comes with a structural twist. At the high level, the agent must decide not only *what* to do (subgoal) but also *how much resource* to spend (budget). Each high-level decision specifies a subgoal together with a budget, and the low level then acts until either the subgoal terminates or the budget is exhausted. This induces variable numbers of primitive steps between high-level decisions, so the high-level process is a *Semi-Markov Decision Process (SMDP)* rather than a standard one-step MDP.

This SMDP structure clashes with many existing goal-conditioned HRL and multi-time-scale world-model methods. Approaches such as Director (Hafner, 2022) and MTS-WM (Shaj Kumar et al., 2023) impose a fixed slow timescale: the high level acts every fixed number of primitive steps, and the world model predicts at that fixed stride. This effectively turns the high level back into an MDP with fixed-duration macro steps. In SSCO, however, the agent must choose *how long* to pursue each subgoal, depending on both the state and the goal. Simply fixing the stride cannot express this. The closest SSCO baseline, WS-option (Feng et al., 2025), does allow variable budgets, but its high level is model-free and emits only a scalar budget. It does not provide semantic subgoals to guide the low level and cannot perform model-based lookahead over variable-duration abstract actions.

We introduce *GoalZero*, a model-based, goal-conditioned HRL framework designed around this SMDP structure. At the high level, GoalZero uses a MuZero-style planner (Schrittwieser et al., 2020) that searches over a set of learned subgoals, while a learned budget head chooses how much resource to allocate to each selected subgoal. To support this planner, we learn an *SMDP-aware world model* that maps a state and a subgoal directly to a post-subgoal latent state and a cumulative reward in a single macro step. Rather than predicting durations explicitly, which can be brittle, we propose a *Multi-Timescale SMDP (MTS-SMDP) world model* in which temporal scale is encoded in the latent geometry: the magnitude of a macro transition reflects how long the corresponding subgoal tends to run. The same latent space also supports low-level, per-step dynamics.

Our main contributions are:

- **Goal-conditioned hierarchical planning in an SMDP.** We formulate the high level of SSCO as a true SMDP, where each decision selects a learned subgoal together with a budget. A MuZero-style planner operates over these subgoals using an SMDP-aware world model, enabling explicit model-based planning over variable-duration abstract actions rather than fixed-stride options or budget-only signals.
- **Multi-timescale SMDP world-model learning via latent geometry.** We propose a unified objective that shapes a single latent space to reflect both micro (primitive-step) and macro (subgoal-level) dynamics. Through complementary "pull", "push", and ranking terms, the learned latent displacements become correlated with the effective durations of subgoals. In the appendix, we show that this geometry–duration relationship emerges from the objective under mild assumptions and relate it to planning regret.
- **Subgoal-conditioned budgets and SSCO performance.** We couple the world model with a subgoal-conditioned budget head, so that the high level decides what strategy to pursue and how much resource to invest in it. On large-scale AIM, SOP, and a Power-2500 benchmark, GoalZero outperforms strong model-free and model-based baselines, highlighting the value of SMDP-aware, multi-timescale planning for stochastic combinatorial optimization.

## 2 RELATED WORKS

**Hierarchical Reinforcement Learning.** HRL methods tackle long-horizon problems by creating temporal abstractions (Barto & Mahadevan, 2003). The options framework (Sutton et al., 1999) and derivatives like Option-Critic (Bacon et al., 2017) formalize temporally extended actions. Recent work learns subgoals either as explicit states with hindsight relabeling (Nachum et al., 2018; Levy et al., 2019) or as learnable continuous representations (Li et al., 2021; Wang et al., 2024). While effective, these are typically *model-free* at the high level, limiting explicit forward planning over SMDP dynamics under stochasticity. WS-option (Feng et al., 2025) is the closest SSCO baseline: it is model-free at the high level (no long-term model-based planning), does not explicitly model SMDP dynamics, and emits only a scalar budget, offering limited semantic guidance to the low level.

**Hierarchical Planning and Search.** Prior work has combined MCTS with temporal abstraction to tackle long horizons. Approaches like Hierarchical MCTS (Vien & Toussaint, 2015) and Option-MCTS (Bai et al., 2016) extend tree search to operate over options, effectively reducing the search depth. However, these methods typically rely on pre-defined options or do not explicitly model the variable-duration dynamics required for resource-constrained SSCO. In the context of subgoal generation, methods like LEAP (Nasiriany et al., 2019) perform search over subgoals to guide a low-level policy. Unlike these approaches, which often assume fixed representations or distance metrics, GoalZero integrates a MuZero-style planner with a learned multi-timescale world model, where the latent geometry itself adapts to the temporal scale of the subgoals.

**Modeling for Semi-Markov Decision Processes.** A key challenge is that the high level operates in an SMDP with variable action durations. Traditional MDP world models assume fixed-duration transitions. Prior work models option duration explicitly via termination learning (Bacon et al., 2017; Khetarpal et al., 2020) or separate duration predictors (Harb et al., 2018), which can be difficult to train and may not capture the interplay between a subgoal's semantics and its execution time. Many goal-conditioned HRL approaches instead assume fixed-length temporal abstractions (Nachum et al., 2018; Sharma et al., 2019; Zhang et al., 2021), simplifying planning but sacrificing flexibility. Our approach learns temporal scale implicitly through latent geometry.

**Unified SMDP dynamics and model-based planning.**   Neural ODEs model continuous-time dynamics over arbitrary intervals (Tzen & Raginsky, 2019), but require costly solvers inside planning loops and do not naturally yield discrete, temporally abstract actions central to HRL. Model-based RL often learns one-step latent dynamics (e.g., Dreamer (Hafner et al., 2020)); when extended hierarchically (Director (Hafner, 2022)), the high level sets latent goals, yet the underlying world model remains one-step. Evaluating a single high-level goal thus requires multi-step latent rollouts of the low-level policy within each node expansion of the search tree. In contrast, our dynamics function $g_\theta$ is trained to predict the *post-subgoal* latent in a single step, collapsing a variable-duration abstract action into one model evaluation. Our MTS-SMDP framework structures the latent space so that transition magnitudes correlate with temporal scale, enabling SMDP-aware planning without an explicit duration predictor. MTS-WM (Shaj Kumar et al., 2023) also models multiple timescales but uses a fixed hyperparameter $H$ to update the slow latent, defining a fixed temporal stride. In contrast, our framework handles agent-chosen, variable durations inherent to SSCO.

**RL for Combinatorial Optimization.**   Applying RL to CO has gained traction (Mazyavkina et al., 2021). Seminal works address TSP (Bello et al., 2017) and VRP (Kool et al., 2019) using Pointer Networks (Vinyals et al., 2015) and Transformers, typically learning one-pass constructive policies for static CO. GoalZero targets the more complex SSCO setting with stochastic transitions and multi-stage resource planning, where hierarchy is essential. Within SSCO, WS-option (Feng et al., 2025) provides an option-based baseline; in contrast, GoalZero enables model-based, goal-conditioned planning with an SMDP-aware world model and subgoal-conditioned budget allocation. SSCO challenges general RL because actions are combinatorial, rewards are non-myopic, and intertemporal budget constraints require variable-duration planning, which fixed-stride HRL methods like Director cannot handle efficiently.

## 3 Preliminaries

**Markov and Semi-Markov Decision Processes**   A standard Markov Decision Process (MDP) is defined by a tuple $\mathcal{M} = (\mathcal{S}, \mathcal{A}, \mathcal{P}, \mathcal{R}, \gamma)$, where $\mathcal{S}$ is the state space, $\mathcal{A}$ is the action space, $\mathcal{P} : \mathcal{S} \times \mathcal{A} \to \Delta(\mathcal{S})$ is the transition probability function, $\mathcal{R} : \mathcal{S} \times \mathcal{A} \to \mathbb{R}$ is the reward function, and $\gamma \in [0, 1]$ is the discount factor. The goal is to find a policy $\pi : \mathcal{S} \to \Delta(\mathcal{A})$ that maximizes the expected cumulative discounted reward $V^\pi(s) = \mathbb{E}_\pi[\sum_{t=0}^{\infty} \gamma^t \mathcal{R}(s_t, a_t)|s_0 = s]$.

Hierarchical reinforcement learning often induces a Semi-Markov Decision Process (SMDP) at the high level. In an SMDP, actions (often called options or subgoals) can take a variable number of primitive timesteps to complete. An SMDP is defined by a tuple $(\mathcal{S}, \mathcal{A}_{HL}, \mathcal{P}_\tau, \mathcal{R}_\tau, \gamma)$, where $\mathcal{A}_{HL}$ is the set of high-level actions. When a high-level action $a_{HL} \in \mathcal{A}_{HL}$ is taken in state $s$, it executes for a stochastic duration $\tau$, drawn from a distribution $p(\tau|s, a_{HL})$. The system transitions to a new state $s'$ according to the transition probability $\mathcal{P}_\tau(s'|s, a_{HL})$, and a cumulative reward $R_\tau(s, a_{HL}) = \mathbb{E}[\sum_{t=0}^{\tau-1} \gamma^t r_t]$ is received. The key challenge is that both the transition dynamics and the rewards are dependent on the variable duration $\tau$.

In the SSCO setting, the high level chooses a **subgoal** $z$ and a **budget** $b$. The low level then acts for a variable number of primitive steps (up to $b$). This induces a random duration $\tau$ between high-level decisions. Unlike standard HRL which often enforces a fixed high-level stride, SSCO requires allocating varying budgets, making the time between decisions dynamic. This invalidates fixed-stride assumptions of methods like Director (Hafner, 2022), necessitating an SMDP formulation.

**Model-Based RL with MCTS**   Model-based RL methods aim to learn a model of the environment's dynamics, $\hat{\mathcal{P}}$ and $\hat{\mathcal{R}}$, which can then be used for planning. Monte Carlo Tree Search (MCTS) is a powerful planning algorithm that explores the decision space by building a search tree. Each node in the tree represents a state, and edges represent actions. MCTS iteratively performs four steps: selection, expansion, simulation (rollout), and backpropagation, to estimate the long-term value of actions from the current state.

**MuZero-style Planning.**   MuZero (Schrittwieser et al., 2020) combines MCTS with learned models in a latent space $\mathcal{Z}$. It consists of:

- A representation function $h_\theta : \mathcal{S} \to \mathcal{H}$ to map an observation to a latent state.
- A dynamics function $g_\theta : \mathcal{H} \times \mathcal{A} \to \mathcal{H} \times \mathcal{R}$ to predict the next latent state $h'$ and reward $r$.
- A prediction function $f_\theta : \mathcal{H} \to \mathcal{P} \times \mathcal{V}$ to predict the policy $\mathbf{p}$ and value $v$.

Planning is performed entirely within this latent space using MCTS to improve the policy target. We adopt this paradigm for our high-level planner.

# 4 THE GOALZERO FRAMEWORK

We present a hierarchical RL framework for SSCO that tackles three *domain-level* challenges: (i) combinatorial action spaces with dynamic feasibility constraints, (ii) stochastic, topology/path-dependent effects that make rewards highly non-myopic, and (iii) intertemporal budget constraints across decision stages. In addition, deploying hierarchy introduces a methodological issue: the high level faces an SMDP with variable-duration subgoals, for which standard one-step world models are ill-suited. GoalZero addresses these with a MuZero-style high-level planner and a multi-timescale SMDP world model, coupled with subgoal-conditioned budget allocation.

## 4.1 HIGH-LEVEL PLANNING WITH MODEL-BASED SEARCH

A central component of our framework is the integration of a MuZero-style model-based planner at the high level, which enables lookahead search in the abstract subgoal space. We provide a detailed description of the network architectures in Appendix F. This design choice is motivated by the unique demands of SSCO. Unlike traditional control problems where actions have immediate, localized effects, decisions in SSCO have long-lasting consequences under uncertainty. A model-based planner allows the agent to "think ahead" by simulating potential sequences of subgoals and their likely outcomes, which is critical for effective resource allocation and strategic positioning.

The high-level planner consists of three learned components, each tailored for SSCO:

1. **Representation function** $h_\theta : \mathcal{S} \to \mathcal{H}$: This function maps the high-dimensional, often graph-structured state $s$ of an SSCO problem into a compact latent representation $h$. For AIM and SOP, we use Graph Neural Networks (GNNs) that can effectively capture the topological structure and node features, producing a fixed-size vector that summarizes the global state of the system. This is crucial for abstracting away the combinatorial complexity of the raw state.
2. **Dynamics function** $g_\theta : \mathcal{H} \times \mathcal{Z} \to (\mathcal{H}, \mathcal{R})$: This function predicts the next latent state $h_{k+1}$ and the cumulative reward $R_k \in \mathcal{R}$ resulting from executing a high-level subgoal, $z_k$, where $\mathcal{R} \subset \mathbb{R}$ is the reward domain. The abstract subgoals themselves are represented as unique, learnable vectors from an embedding layer, allowing their semantic meaning to be discovered end-to-end during training. The predicted reward, $R_k = \sum_{t=0}^{\tau_k - 1} \gamma^t r_t$, represents the total discounted reward over the subgoal's variable duration $\tau_k$. By learning to predict the outcome of this entire sequence of low-level actions in a single step, rather than one primitive action at a time, the dynamics function creates a temporal abstraction that is critical for efficient, long-horizon planning.
3. **Prediction function** $f_\theta : \mathcal{H} \to \mathcal{P} \times \mathcal{V}$: From a latent state $h_k$, this function outputs two crucial estimates:
   - A **policy** $\mathbf{p}_k \in \mathcal{P}$, which is a prior probability distribution over the set of high-level subgoals $\mathcal{Z}$.
   - A **value** $v_k \in \mathcal{V}$, which is a scalar estimate of the expected future cumulative reward from that state, where $\mathcal{V} \subset \mathbb{R}$ is the value domain.

   Crucially, this predicted value $v_k$ represents the long-term return, distinct from the immediate, one-step reward $R_k$ predicted by the dynamics function. These predictions guide the MCTS search, focusing the planning effort on promising subgoals and allowing for efficient evaluation of states without needing to perform a full rollout to the end of the episode.

**Planning Process.** As illustrated in Fig. 1, the high-level action is a tuple $a_{HL} = (z, b)$, where $z \in \mathcal{Z}$ is a discrete subgoal and $b \in \{0, \ldots, B_{\max}\}$ is the budget, so that $\mathcal{A}_{HL} = \mathcal{Z} \times \mathcal{B}$. The planner operates as follows: 1. Encode state $s_k$ to $h_k = h_\theta(s_k)$. 2. Run MCTS using $g_\theta$ and $f_\theta$ over subgoals $z$. 3. Obtain MCTS policy $\pi_k$ and sample subgoal $z_k \sim \pi_k$. 4. Sample budget $b_k$ from the budget head $b_\psi(\cdot | h_k, z_k)$.

## 4.2 UNIFIED MULTI-TIMESCALE WORLD MODEL LEARNING

A central difficulty in our setting is that the high-level policy lives in an SMDP: executing a subgoal and its budget from a state leads to a stochastic, variable number of primitive steps before the next high-level decision. We would like a *single* world model that supports both (i) micro-scale MDP dynamics at the level of primitive steps and (ii) macro-scale dynamics at the level of subgoals that

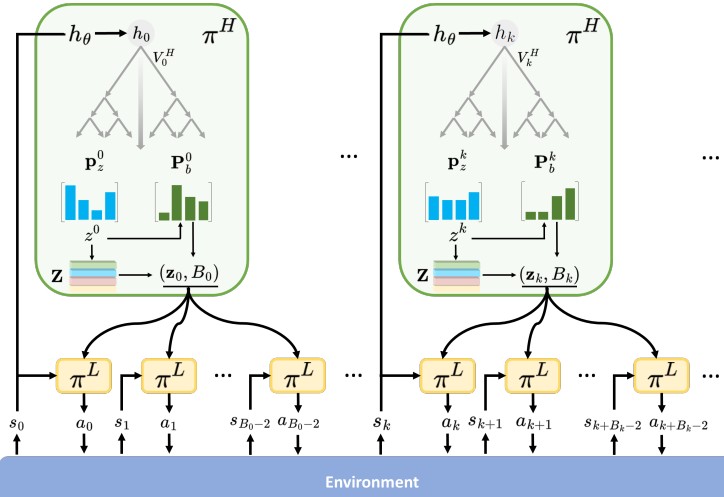

Figure 1: Architecture of GoalZero. At each high-level (HL) decision step, the MuZero-style planner encodes the current state with the representation network $h_\theta$, runs tree search in latent space, and outputs a distribution over subgoals $\mathbf{p}_z^k$ and a subgoal-conditioned budget distribution $\mathbf{p}_b^k$. A high-level action is the pair $(z_k, b_k)$. The low-level (LL) policy $\pi^L$ then executes primitive actions in the environment to pursue subgoal $z_k$ within budget $b_k$. The planner comprises a representation network $h_\theta$, a dynamics network $g_\theta$, and a prediction network $f_\theta$. These components are trained jointly with our unified multi-timescale SMDP objective, which shapes the latent geometry so that the magnitude of $g_\theta$'s macro-transitions reflects the temporal scale of subgoals.

may run for many steps. In addition, this model should be *planning ready*: given a latent state and a candidate subgoal, the high-level planner should be able to predict the post-subgoal outcome in one model evaluation, without explicitly rolling out the low-level policy inside the search.

Our approach is to learn a *unified latent space* in which temporal scale is encoded geometrically. The dynamics network $g_\theta$ implements a one-step SMDP transition

$$g_\theta : \mathcal{H} \times \mathcal{Z} \to \mathcal{H} \times \mathcal{R}, \qquad g_\theta(h_\theta(s), z) = \big(h'_\theta(s, z), R_\theta(s, z)\big),$$

where $h'_\theta(s, z)$ summarizes the post-subgoal latent state and $R_\theta(s, z)$ is the cumulative reward over the (random) subgoal duration. Let $g_\theta^h(h, z)$ denote the latent-state component of this output. We define the latent distance of a macro transition as

$$d_\theta(s, z) \triangleq d\big(h_\theta(s), g_\theta^h(h_\theta(s), z)\big),$$

where $d(\cdot, \cdot)$ is the chordal distance between unit-normalized latents on the sphere, and we score a subgoal by

$$f_{\theta, \phi}(s, z) = d_\theta(s, z) + m_\phi(z),$$

with $m_\phi(z)$ a learned margin head that is regularized to avoid unbounded growth. In this design, the *size* of the macro move encodes the temporal scale associated with $z$, while the shared encoder $h_\theta$ also supports per-step micro dynamics.

We structure the training objective around three forces, inspired by attractive and repulsive dynamics in contrastive learning (Hadsell et al., 2006):

1. **Micro-scale Pull.** We need a consistent "ruler" for the latent space. A *Low-Level Cap* penalizes latent jumps between consecutive primitive states that exceed a small threshold, so that one primitive step corresponds to a bounded latent distance.
2. **Macro-scale Push.** We need subgoals to represent temporally extended behaviors. A *High-Level Margin* enforces that the latent displacement produced by a macro transition for subgoal $z$ is at least a learned margin $m_\phi(z)$, preventing long-duration subgoals from collapsing to negligible moves.
3. **Temporal Order.** To make the planner trust these distances, we explicitly align the score $f_{\theta, \phi}(s, z)$ with the *effective duration* of subgoals via a budget-aware ranking loss over pairs of executed subgoals.

Together, these terms shape a *multi-timescale* geometry: micro dynamics are locally smooth, while macro transitions reflect variable, agent-chosen temporal scopes.

**Formal objective.** For a pair of executed high-level actions $(z_i, b_i)$ and $(z_j, b_j)$ from the same state $s$, let $Y \in \{-1, 0, 1\}$ indicate which execution took longer (ties are dropped; see Appendix B.1). We optimize three complementary losses that encourage (i) low-level MDP consistency, (ii) high-level under-displacement, and (iii) budget-aware order, respectively. For clarity we list the constituent terms below and defer the full training objective to Eq. (5). We use $(x)_+ \triangleq \max\{0, x\}$.

$$\mathcal{L}_{\mathrm{cap}}(\theta) := \underbrace{\mathbb{E}_{(s_t, s_{t+1} | z, b)} \big[ \big( d(h_\theta(s_t), h_\theta(s_{t+1})) - \kappa \big)_+^2 \big]}_{\text{Low-Level MDP Consistency (\textit{Pull})}}, \tag{1a}$$

$$\mathcal{L}_{\mathrm{push}}(\theta, \phi) := \underbrace{\mathbb{E}_{(s, z)} \big[ \big( m_\phi(z) - d_\theta(s, z) \big)_+ \big]}_{\text{High-Level Under-Displacement (\textit{Push})}}, \tag{1b}$$

$$\mathcal{L}_{\mathrm{order}}(\theta, \phi) := \underbrace{\mathbb{E}_{(s; z_i, b_i, z_j, b_j)} \big[ \big( 1 - Y \left( f_{\theta, \phi}(s, z_i) - f_{\theta, \phi}(s, z_j) \right) \big)_+ \big]}_{\text{Budget-Aware Order Consistency}}. \tag{1c}$$

This unified objective is designed to impose a meaningful temporal structure on the latent space. The **Low-Level MDP Consistency** term acts as a geometric regularizer, controlling the expected per-step latent motion (the "pull"). As derived in **Lemma B.6** (Appendix B), minimizing this loss allows us to upper-bound the expected total latent path length of a low-level trajectory by roughly proportional to its expected duration $\mathbb{E}[\tau]$. This establishes the foundational link between geometry and time.

The key insight is how these geometric constraints force temporal awareness to emerge. The **High-Level Under-Displacement** term encourages the latent displacement $d_\theta(s, z)$ to meet or exceed the learned margin $m_\phi(z)$ (the "push"). By combining this lower bound on displacement with the upper bound from the cap, **Proposition B.7** proves that the learned margin $m_\phi(z)$ is forced to become a principled lower bound for the subgoal's **expected duration** $\mathbb{E}[\tau \mid s, z]$. The planner can therefore use this geometric quantity as a direct, learnable proxy for temporal cost, without introducing a separate *duration-prediction head in the world model*.

Finally, the **Budget-Aware Order Consistency** term ensures this learned temporal structure is correctly calibrated for decision-making. We prove in **Theorem B.8** that minimizing this ranking loss yields a model scorer, $f_{\theta, \phi}$, that orders subgoals consistently with their effective, budget-aware mean durations. The full theoretical analysis in Appendix B confirms that this structure is sound for planning, culminating in a regret bound (**Theorem B.9**) that connects the quality of the learned components directly to planning performance over high-level macro-actions.

**Relation to prior HRL and multi-time-scale models.** This unified objective yields what we call a *Multi-Timescale SMDP world model*: a single latent space simultaneously respects micro-scale MDP dynamics and macro-scale SMDP dynamics, with temporal scale emerging from geometry rather than from a fixed slow stride. This contrasts with:

- **Model-free options and budget-only HRL** (*e.g.*, WS-option), which learn termination or budgets but do not provide a planning-ready SMDP model and communicate only "how long" to act without semantic subgoals.
- **Fixed-stride hierarchical world models** (*e.g.*, Director, MTS-WM), where slow dynamics are defined at a fixed interval, so the high level effectively operates with fixed-duration macro steps. In contrast, our macro transitions compress a *variable* number of micro steps into one latent move, and their length adapts to the subgoal and environment.

In the appendix, we show that under primitive assumptions, this objective suffices to derive a geometry–duration coupling, a duration lower bound from the learned margins, and an order-consistent scorer, and we relate these properties to planning regret (Theorems B.6 to B.9). Empirically, we find that this multi-timescale geometry is essential for making MuZero-style high-level planning effective in SSCO.

### 4.3 Subgoal-Conditioned Budget Allocation

SSCO problems involve budget constraints that must be allocated across sequential decisions. We propose a subgoal-conditioned budget head that learns to predict resource allocation in the context of

our multi-timescale framework:

$$b_\psi : \mathcal{H} \times \mathcal{A}_{HL} \to \Delta_B \qquad (2)$$

where $\Delta_B = \{0, 1, ..., B_{\text{remaining}}\}$ denotes the discrete budget space. This learned allocation is conditioned on both the current latent state and the selected subgoal, facilitating context-aware resource distribution that can reflect each subgoal's temporal scale.

**Budget head as a learned policy (actor–critic).** Following (Feng et al., 2025), we treat the budget head $b_\psi(b \mid s, z)$ as a policy over discrete budgets $b \in \{0, \ldots, B_{\max}\}$ conditioned on the HL state and subgoal. Let $G_t$ denote the return from the HL step (search-backed value targets as in MuZero). We maximize

$$\mathcal{J}(\psi) = \mathbb{E}\left[ \sum_t \log b_\psi(b_t \mid s_t, z_t) A_t \ + \ \beta\, \mathcal{H}(b_\psi(\cdot \mid s_t, z_t)) \right],$$

with advantage $A_t = G_t - V_\eta(s_t, z_t)$ from a learned critic $V_\eta$ and entropy regularizer $\beta > 0$. Gradients are $\nabla_\psi \mathcal{J} = \mathbb{E}\left[ \sum_t \nabla_\psi \log b_\psi(b_t \mid s_t, z_t) A_t + \beta \nabla_\psi \mathcal{H} \right]$.

### 4.4 Low-Level Policy for Combinatorial Actions

Once the high-level planner selects a strategic subgoal $z_k$ and allocates a corresponding budget $b_k$, the low-level policy translates this directive into a concrete sequence of primitive combinatorial actions. At this level, the agent faces the classic SSCO challenge: selecting from a vast, discrete action space under local information. A purely myopic policy would fail to align its actions with the long-term plan.

Conditioning on $z_k$ is therefore important. The subgoal embedding communicates strategic intent and implicit temporal scale learned by the MTS-SMDP world model, guiding the low-level policy on *what* to achieve. We implement the low-level policy as a subgoal-conditioned action-value network trained with a DQN objective. A GNN processes the current graph state, concatenating $z_k$ to each node's representation to compute Q-values for each primitive action (*e.g.*, node selection) in the context of the overarching strategy.

The network minimizes the Bellman error over transitions in a low-level replay buffer $\mathcal{D}_{LL}$:

$$\mathcal{L}_{LL} = \mathbb{E}_{(s_t, z_k, a_t, r_t, s_{t+1}) \sim \mathcal{D}_{LL}} \left[ \left( Q(s_t, a_t \mid z_k) - y_t \right)^2 \right] \qquad (3)$$

with targets computed using a target network $Q'$:

$$y_t = r_t + \gamma \max_{a'} Q'(s_{t+1}, a' \mid z_k). \qquad (4)$$

During execution, the low-level policy uses this action-value function to generate primitive actions that fulfill the subgoal, acting for the budgeted duration $b_k$ and translating the high-level plan into a concrete combinatorial solution.

## 5 Experiments

### 5.1 Experimental Setup

**Domains** We evaluate our framework on two canonical SSCO domains, now on larger and more challenging problem instances than those used in prior work. To ensure a direct and fair comparison, the environments and problem parameters are scaled up but remain consistent in principle with Feng et al. (2025)[1].

- **Adaptive Influence Maximization (AIM):** A stochastic propagation problem on graphs, for which we use instances with $N = 500$ nodes.
- **Stochastic Orienteering Problem (SOP):** A stochastic route planning problem, for which we test on instances with $N = 500$ nodes.

**Evaluation Protocol** To ensure a robust assessment, we evaluate all methods across a matrix of problem settings, varying both the episode horizon $(T)$ and the total resource budget $(K)$. Specifically, we test on configurations of $(T, K) \in \{(10, 50), (10, 60), (10, 70), (20, 10)\}$.

---

[1] Full implementation details for both environments are provided in Appendix E.

**Statistical reporting.** We run 10 random seeds per configuration and report **mean $\pm$ standard error of the mean (s.e.m.).** $p$-values are computed from two-sided Welch $t$-tests on seed means.

**Baselines** We compare GoalZero against a comprehensive suite of baselines, including those from the most closely related work, WS-option (Feng et al., 2025), as well as our own ablations. In addition, we compare against two general-purpose model-based RL baseline, *i.e.*, Director-style GC-HRL (Hafner, 2022) and a Flat MuZero agent (Schrittwieser et al., 2020), with detailed results reported in Appendix L.2.

For the AIM problem, we adopt the baselines from Tong et al. (2020); Feng et al. (2025), structured as 'HL-Policy - LL-Policy':

- **High-Level Budget Policies:**
  - `average`: Divides the total budget $K$ evenly across the $T$ stages.
  - `normal`: Allocates the entire budget $K$ in the first stage.
  - `static`: Divides the horizon into cycles and allocates budget only at the start of each cycle.
- **Low-Level Node Selection Policies:**
  - `degree`: A greedy heuristic selecting nodes with the highest out-degree.
  - `score`: A greedy heuristic selecting nodes based on their expected immediate influence, $s_v = \sum_{u \in \mathcal{N}_{\text{in}}(v)} p(u, v)$.
- **Flat DQN:** A non-hierarchical RL baseline. This agent uses a single GNN-based Q-network to select one node at a time, up to the total budget $K$. It represents a standard deep RL approach without temporal abstraction.

For the SOP problem, we compare against standard optimization baselines:

- **Greedy Heuristic:** A domain-specific myopic policy that greedily selects the highest-profit node reachable within the daily distance limit.
- **Genetic Algorithm (GA):** A strong meta-heuristic baseline that optimizes a fixed sequence of daily budget allocations over the entire horizon.
- **Flat DQN:** A non-hierarchical RL baseline, analogous to the one used for AIM, which makes a single routing decision at each primitive step.

## 5.2 COMPARISON WITH BASELINES

As shown in Tables 1 and 2, GoalZero outperformed the included heuristic and learning-based baselines across the tested configurations. We observed the performance gap widen as the problem complexity (*e.g.*, budget $K$) increases, highlighting the strength of GoalZero's long-horizon planning capabilities enabled by the multi-timescale world model.

The learning curves in Figure 2 further illustrate the advantages of our approach: GoalZero not only converges to a higher final performance but also shows greater sample efficiency and stability throughout training, whereas WS-option often exhibits larger variance and slower convergence, especially in the SOP domain, which is consistent with the difficulty of credit assignment for model-free high-level policies in SMDPs. The Flat DQN baseline sometimes learns quickly yet consistently plateaus at a suboptimal level, highlighting the importance of hierarchical abstraction for effective long-horizon planning in these complex settings. Although WS-option is a strong baseline in AIM, its performance degrades markedly on SOP, exposing a limitation of budget-only HRL in tasks requiring spatial reasoning: SOP is essentially a routing problem where the value of an action depends on the global path, so a scalar budget conveys *how long* to travel but not *where* to go, leaving the low-level policy to explore myopically. In contrast, GoalZero's subgoals can capture *target regions* or *structural patterns*, *e.g.*, community clusters in AIM (Fig. 3a) and spatially dense neighborhoods in SOP (Fig. 3b, dashed circle), providing the spatial context that, in our analysis, helps the MCTS planner assemble globally coherent routes.

## 5.3 ASSESSING THE LEARNED POLICIES

To rigorously dissect the contributions of GoalZero's core architectural components, we conducted a comprehensive ablation study. We progressively build up from an option-based baseline (A) to the full GoalZero model (E), quantifying the performance gain at each step. The results, presented in Table 3 for both the AIM ($N = 500, K = 70$) and SOP ($N = 500, K = 70$) domains, support our design choices and highlight the synergy between the framework's components.

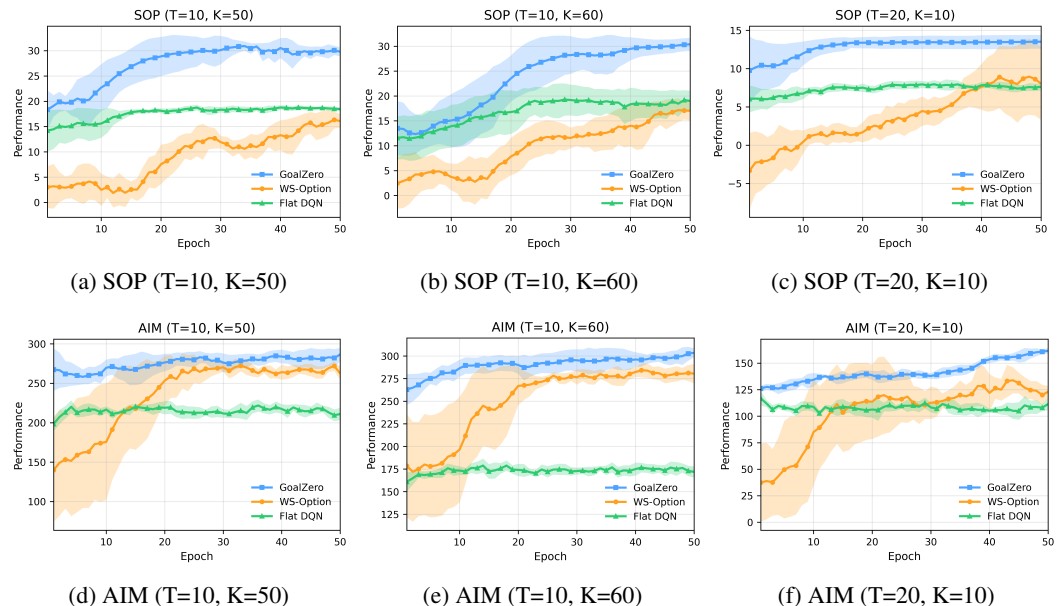

Figure 2: Training curves comparing GoalZero against learning-based baselines across various AIM and SOP settings. The solid line denotes the mean performance over 10 seeds, and the shaded area represents the standard error of the mean. GoalZero consistently demonstrates superior sample efficiency and converges to a higher and more stable final performance.

Table 1: Experimental results for AIM on $N = 500$ graphs. GoalZero demonstrates superior performance, especially in settings with larger budgets where long-term planning is more critical. All improvements of GoalZero over WS-option are statistically significant (p-value $\leq 0.05$).

| Method | $N = 500$ | | | |
|---|---|---|---|---|
| | $T, K = 10, 50$ | $T, K = 10, 60$ | $T, K = 10, 70$ | $T, K = 20, 10$ |
| **GoalZero (Ours)** | **286.11** $\pm$**4.05** | **303.50** $\pm$**2.52** | **324.15** $\pm$**2.38** | **161.71** $\pm$**1.39** |
| WS-option | 268.23 $\pm$3.75 | 280.44 $\pm$5.46 | 301.53 $\pm$9.10 | 123.02 $\pm$2.88 |
| Flat DQN | 255.56 $\pm$4.15 | 171.80 $\pm$3.16 | 243.46 $\pm$3.78 | 111.46 $\pm$2.95 |
| average-degree | 268.16 $\pm$4.29 | 292.83 $\pm$4.68 | 311.35 $\pm$3.91 | 137.26 $\pm$2.71 |
| average-score | 272.26 $\pm$4.38 | 293.81 $\pm$4.75 | 313.74 $\pm$3.02 | 150.36 $\pm$3.01 |
| normal-degree | 127.27 $\pm$2.43 | 147.09 $\pm$2.81 | 165.82 $\pm$2.15 | 31.41 $\pm$1.05 |
| normal-score | 134.31 $\pm$2.59 | 155.74 $\pm$2.99 | 172.95 $\pm$2.33 | 34.43 $\pm$1.12 |
| static-degree | 266.51 $\pm$4.27 | 287.66 $\pm$4.61 | 296.18 $\pm$3.95 | 117.26 $\pm$2.33 |
| static-score | 269.21 $\pm$4.31 | 291.01 $\pm$4.69 | 310.82 $\pm$4.00 | 130.09 $\pm$2.65 |

Table 2: Experimental results for SOP on $N = 500$ graphs. GoalZero's ability to plan under uncertainty yields substantial gains over myopic and meta-heuristic baselines. All improvements of GoalZero over WS-option are statistically significant (p-value $\leq 0.05$).

| Method | $N = 500$ | | | |
|---|---|---|---|---|
| | $T, K = 10, 50$ | $T, K = 10, 60$ | $T, K = 10, 70$ | $T, K = 20, 10$ |
| **GoalZero (Ours)** | **29.79** $\pm$**0.46** | **30.39** $\pm$**0.67** | **31.60** $\pm$**1.94** | **13.51** $\pm$**0.45** |
| WS-option | 16.08 $\pm$1.32 | 17.03 $\pm$0.99 | 12.99 $\pm$1.25 | 8.04 $\pm$2.41 |
| Flat DQN | 18.44 $\pm$0.18 | 18.99 $\pm$1.06 | 15.90 $\pm$0.67 | 7.62 $\pm$0.22 |
| GA | 12.35 $\pm$0.21 | 14.51 $\pm$0.27 | 15.23 $\pm$0.22 | 7.91 $\pm$0.18 |
| Greedy | 17.17 $\pm$0.22 | 21.91 $\pm$0.25 | 23.58 $\pm$0.16 | 4.89 $\pm$0.15 |

**From Budget-Only to Goal-Conditioned Execution** Our starting point is a strong option HRL baseline akin to *WS-option* (A), which uses a model-free high-level policy to allocate a budget but does not communicate a strategic subgoal. In our next variant (B), we introduce a MuZero planner that generates subgoals and conditions the low-level policy on them, while keeping the budget allocation independent of the subgoal. The performance gain from A to B demonstrates the immediate value of

| (a) AIM Ablation ($N = 500, K = 70$). | |
|---|---|
| **Algorithm Variant** | **Avg. Reward** |
| A: Baseline | 301.53 $\pm$9.10 |
| B: + LL Subgoal-Cond. | 309.52 $\pm$3.71 |
| C: + Budget Subgoal-Cond. | 314.65 $\pm$3.80 |
| D: + MTS (Fixed Margin) | 322.04 $\pm$2.91 |
| **E: GoalZero (Full Model)** | **324.15 $\pm$2.38** |
| F: GoalZero-MF (Model-Free) | 307.29 $\pm$5.62 |

| (b) SOP Ablation ($N = 500, K = 70$). | |
|---|---|
| **Algorithm Variant** | **Avg. Reward** |
| A: Baseline | 12.99 $\pm$1.25 |
| B: + LL Subgoal-Cond. | 17.03 $\pm$1.98 |
| C: + Budget Subgoal-Cond. | 22.75 $\pm$1.05 |
| D: + MTS (Fixed Margin) | 28.62 $\pm$1.19 |
| **E: GoalZero (Full Model)** | **31.60 $\pm$1.94** |
| F: GoalZero-MF (Model-Free) | 15.89 $\pm$1.91 |

Table 3: Ablation study results. The progressive addition of GoalZero's core components leads to consistent performance improvements across both domains.

providing the low-level policy with strategic direction (*what* to do), even when the resource allocation (*how long* to act) is not yet aligned with that strategy. Note that at this stage, the MuZero planner is trained without the MTS objective.

**Aligning Resources with Strategy**    In variant C, we take the logical next step by also conditioning the budget allocation on the selected subgoal. The performance improvement from B to C shows the benefit of this alignment. When the agent can allocate more resources to more ambitious subgoals and fewer to simple ones, its planning becomes more efficient and effective. At this stage, we have a complete goal-conditioned HRL agent, but its world model remains a standard predictive model.

**The Critical Role of the MTS World Model**    A considerable performance improvement occurs when moving from variant C to D, where we introduce the *MTS objective with a fixed margin*. For this, we use the average budget times of the low level fixed margin $\kappa$. This variant uses our proposed push-pull dynamic to structure the latent space, but with a single, fixed temporal scale for all subgoals. This improvement underscores our central claim: a generic application of model-based planning may be insufficient for SMDPs. Explicitly structuring the latent space for multi-timescale reasoning is the critical component that unlocks effective planning over temporally abstract actions.

Moving from variant D to the full GoalZero model (E), we replace the fixed margin with the proposed dynamic, learnable margin predictor. The resulting performance gain highlights the value of adaptive learning strategies. Allowing the agent to discover that different subgoals have different intrinsic temporal scales provides the final layer of strategic flexibility required to master these complex, multi-stage optimization problems.

**The Benefit of High-Level Planning**    Finally, to isolate the contribution of model-based planning itself, we evaluate *GoalZero-MF (Model-Free)* (F). This variant retains the full hierarchical and goal-conditioned structure of GoalZero but replaces the MuZero planner with a reactive, model-free DQN policy. Its performance is clearly inferior to the full, planning-based model (E). This result confirms that in stochastic combinatorial domains, the foresight provided by the model-based planner is a key driver of performance, enabling the agent to find more robust and effective long-term strategies.

## 6    CONCLUSION

We introduced GoalZero, a model-based hierarchical reinforcement learning framework designed for Sequential Stochastic Combinatorial Optimization. Its core contribution is a novel method for Multi-Timescale SMDP World Model Learning, which enables a MuZero-style planner to learn and reason over adaptive temporal abstractions. By training a dynamics model to produce latent transitions whose magnitudes correspond to learned, subgoal-specific temporal scales, our framework sidesteps the need for explicit duration modeling and allows for the natural emergence of diverse, multi-horizon strategies. Complemented by an integrated, subgoal-conditioned budget allocation mechanism, GoalZero achieved improved long-horizon performance in our experiments. We demonstrate empirically that GoalZero outperforms strong baselines on complex SSCO problems. Our analysis suggests that its ability to learn and plan over adaptive temporal abstractions is a critical factor in its performance, highlighting a promising direction for applying HRL to other complex planning domains.

## REPRODUCIBILITY STATEMENT

We took several steps to support reproducibility. The method is specified in Sec. 4, including the unified objective in Eq. (5) and the training/inference loop summarized in Algorithm D. Experimental settings (domains, $(T, K)$ configurations, and evaluation protocol) are given in Sec. 5, where we report mean $\pm$ s.e.m. over 10 seeds and compute $p$-values using two-sided Welch $t$-tests on seed means. Environment generation and preprocessing details for AIM and SOP are provided in Appendix E. Network architectures and heads appear in Appendix F, with hyperparameters consolidated in Table 4; budget actor–critic training details are in Appendix G. We include ablations in Appendix H, hyperparameter sensitivity in Appendix I, large-graph generalization in Appendix J, and Kendall's $\tau$ validation in Appendix L. Theoretical assumptions and statements are collected in Appendix B, with complete proofs in Appendix B. An anonymous link to the source code is provided in Appendix F.

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

APPENDIX CONTENTS

## A  THE USE OF LARGE LANGUAGE MODELS (LLMS)

We used an LLM strictly as a general-purpose writing assistant to aid with language clarity and polishing. Concretely, it was employed to (i) rephrase sentences for readability, (ii) improve grammar and style consistency, and (iii) help with minor LaTeX boilerplate such as table captions and section cross-references. The LLM did *not* contribute to research ideation, algorithm design, implementation, data generation, experimental setup, hyperparameter selection, or analysis. All technical content including the formulation of the unified objective, algorithmic choices, proofs, experiments, figures/tables, and conclusions, was conceived, implemented, and validated by the authors. Any LLM-suggested wording was manually reviewed for accuracy and faithful intent before inclusion. No code, proofs, or empirical results were produced by the LLM.

## B  THEORETICAL ANALYSIS FOR THE MTS-SMDP WORLD MODEL

This appendix provides complete assumptions and detailed theoretical analysis. We refer to Eq. (5) in the main text for the unified objective and to Appendix B.1 for compact notation. All latent outputs of $h_\theta$ and $g_\theta$ are $\ell_2$-normalized to lie on the unit sphere $\mathbb{S}^{d-1}$, and $d(\cdot, \cdot)$ denotes the chordal distance on the unit sphere, as defined in the main text. We use $\mathbf{1}\{\cdot\}$ for indicator functions and $\{\mathcal{F}_i\}_{i \geq 0}$ for the filtration up to step $i$.

### B.1  NOTATION AND UNIFIED LOSS

Let $h_\theta : \mathcal{S} \to \mathbb{R}^d$ and $g_\theta : \mathbb{R}^d \times \mathcal{Z} \to \mathbb{R}^d$ be the shared encoder and HL transition heads; both outputs are $\ell_2$-normalized on forward pass to lie on $\mathbb{S}^{d-1}$. We write the chordal distance on the unit sphere as $d(\cdot, \cdot)$ (defined in the main text). Define the macro-step latent displacement

$$d_\theta(s, z) = d\big(h_\theta(s),\, g_\theta(h_\theta(s), z)\big), \qquad f_{\theta,\phi}(s, z) = d_\theta(s, z) + m_\phi(z),$$

where $m_\phi$ is the learnable margin head (with an $\ell_2$ regularizer). Let $(z, b)$ denote the chosen HL subgoal and budget; executing $(z, b)$ from $s$ induces a random path $(s_0{=}s, s_1, \ldots, s_\tau)$ with realized duration $\tau \in \{0, \ldots, b\}$. For pairs $(z_i, b_i)$, $(z_j, b_j)$ we store labels $Y = \mathrm{sign}\big(\tau_{b_i}(s, z_i) - \tau_{b_j}(s, z_j)\big) \in \{-1, 0, 1\}$.

We train with the unified loss:

$$\begin{aligned}
\mathcal{L}_{\mathrm{unified}}(\theta, \phi) = {}& w_{\mathrm{cap}} \underbrace{\mathbb{E}_{(s_t, s_{t+1}|z, b)}\Big[\big(d\big(h_\theta(s_t), h_\theta(s_{t+1})\big) - \kappa\big)_+^2\Big]}_{\text{Low-Level MDP Consistency (\textit{Pull})}} \\
&+ w_{\mathrm{push}} \underbrace{\mathbb{E}_{(s, z)}\big[\big(m_\phi(z) - d_\theta(s, z)\big)_+\big]}_{\text{High-Level Under-Displacement (\textit{Push})}} \\
&+ w_{\mathrm{order}} \underbrace{\mathbb{E}_{(s; z_i, b_i, z_j, b_j)}\Big[\big(1 - Y\big(f_{\theta,\phi}(s, z_i) - f_{\theta,\phi}(s, z_j)\big)\big)_+\Big]}_{\text{Budget-Aware Order Consistency}} + \lambda_m \|m_\phi\|_2^2,
\end{aligned} \tag{5}$$

where $(x)_+ = \max\{0, x\}$ and $\kappa > 0$ is the LL per-step cap on latent motion (unit sphere). Unless otherwise specified, experiments use $w_{\mathrm{cap}} = w_{\mathrm{push}} = w_{\mathrm{order}} = 0.10$.

### B.2  ASSUMPTIONS

We rely on the following assumptions. Crucially, we do **not** assume a priori that the learned geometry correlates with duration; rather, we assume standard boundedness and regularity conditions, and subsequently *derive* the geometry-duration coupling as a consequence of minimizing the objective in Eq. (5).

**Assumption B.1** (Bounded Expected Modeling Error). There exists $\bar{\varepsilon} \geq 0$ such that for any $(s, z)$,

$$\mathbb{E}\big[d\big(g_\theta(h_\theta(s), z),\, h_\theta(s')\big) \mid s, z\big] \leq \bar{\varepsilon},$$

where $s'$ is the terminal state reached by executing the macro-action $(s, z)$ in the environment.

**Assumption B.2** (Cap Exceedance Budget (Conditional)). Let $d_i d\big(h_\theta(s_{i+1}), h_\theta(s_i)\big)$ be the per-step latent distance along the low-level rollout that realizes the macro-action $(s, z)$, and define $X_i(d_i - \kappa)_+$. Let $\{\mathcal{F}_i\}_{i \geq 0}$ be the canonical filtration generated by $(s_0 = s, s_1, \ldots, s_i)$ and any internal randomness up to step $i$. There exists $\xi \geq 0$ such that

$$\mathbb{E}[X_i \mid \mathcal{F}_i] \leq \xi \quad \text{a.s. for all } i \geq 0.$$

**Assumption B.3** (Informative Pairwise Labels). Let $\bar{\mu}(s, z) \mathbb{E}_{b \sim \pi_b}[\mathbb{E}[\tau_b \mid s, z]]$ denote the effective mean duration under the budget distribution $\pi_b$. For any informative pair $(s; z_i, z_j)$ with $\bar{\mu}(s, z_i) \neq \bar{\mu}(s, z_j)$,

$$\Pr\{Y = 1 \mid s; z_i, z_j\} > \tfrac{1}{2} \quad \Longleftrightarrow \quad \bar{\mu}(s, z_i) > \bar{\mu}(s, z_j),$$

where $Y = \text{sign}\big(\tau_{b_i}(s, z_i) - \tau_{b_j}(s, z_j)\big) \in \{-1, 0, 1\}$.

**Assumption B.4** (Standard Regularity and Value Smoothness). (i) Rewards are bounded $r_t \in [0, R_{\max}]$, with discount $\gamma \in [0, 1)$.
(ii) $\tau$ has finite variance.
(iii) The state-value (or macro-action value) is Lipschitz w.r.t. effective duration with constant $L_V = \frac{R_{\max}}{1 - \gamma}$,

$$|V(s, z_i) - V(s, z_j)| \leq L_V \, |\bar{\mu}(s, z_i) - \bar{\mu}(s, z_j)|.$$

(iv) The value is also Lipschitz in the latent geometry: there exists $L_S \geq 0$ such that for any two latent states $u, v \in \mathbb{S}^{d-1}$,

$$|V(u) - V(v)| \leq L_S \, d(u, v).$$

Assumption B.2 matches the squared-hinge cap penalty in Eq. (5) (Pull term) and is stated conditionally, which allows us to control sums up to the stopping time $\tau$ without invoking Wald's identity.

*Remark* B.5 (Justification for Lipschitz Value). Assumption B.4(iv) postulates that the value function varies smoothly with respect to the latent geometry. While strong, this assumption is motivated by two factors. First, from an architectural standpoint, the value function is parameterized by a neural network $f_\theta$ (the prediction head); given bounded weights (enforced via $\ell_2$ regularization) and Lipschitz activation functions (e.g., ReLU), the function is Lipschitz continuous by construction (Gouk et al., 2021). Second, the training objective explicitly minimizes value prediction error; assuming the representation learning is successful, the encoder $h_\theta$ will map states with similar values to nearby points in the latent space to minimize this loss, thereby inducing the smoothness described in the assumption.

## B.3 DERIVING MACRO–MICRO CONTROL FROM THE OBJECTIVE

We first show that minimizing the **Low-Level MDP Consistency (Pull)** term imposes an upper bound on the latent displacement proportional to the execution duration. This is a geometric consequence of the triangle inequality and the cap loss.

**Lemma B.6** (Expectation-Level Macro–Micro Control). *Under Assumptions B.1 and B.2, for any $(s, z)$ we have*

$$\mathbb{E}[d(h_\theta(s), h_\theta(s')) \mid s, z] \leq \mathbb{E}\Big[\sum_{i=0}^{\tau-1} d\big(h_\theta(s_{i+1}), h_\theta(s_i)\big) \Big| s, z\Big] \leq \mathbb{E}[\tau \mid s, z] \, (\kappa + \xi), \quad (6)$$

*and furthermore*

$$\big| \mathbb{E}[d_\theta(s, z) \mid s, z] - \mathbb{E}[d(h_\theta(s), h_\theta(s')) \mid s, z] \big| \leq \bar{\varepsilon}. \quad (7)$$

*Proof.* Fix $(s, z)$. For the left inequality in equation 6, consider a realized low-level path $(s_0 = s, s_1, \ldots, s_\tau = s')$. Since $d$ is a metric (chordal distance), the triangle inequality implies:

$$d\big(h_\theta(s), h_\theta(s')\big) \leq \sum_{i=0}^{\tau-1} d\big(h_\theta(s_i), h_\theta(s_{i+1})\big).$$

Taking conditional expectation given $(s, z)$ yields the first inequality.

For the right inequality in equation 6, define $d_i d\big(h_\theta(s_{i+1}), h_\theta(s_i)\big)$ and the filtration $\{\mathcal{F}_i\}$ generated by the rollout up to $s_i$ (including any internal randomness). Note that for each $i$,

$$d_i \leq \kappa + (d_i - \kappa)_+,$$

hence by Assumption B.2 (which holds when the Cap loss is minimized to level $\xi$),

$$\mathbb{E}[\,d_i \mid \mathcal{F}_i\,] \;\leq\; \kappa + \mathbb{E}[(d_i - \kappa)_+ \mid \mathcal{F}_i] \;\leq\; \kappa + \xi \quad \text{a.s.}$$

Since $d_i \mathbf{1}\{i < \tau\} \geq 0$, Tonelli's theorem allows us to exchange the sum and expectation:

$$\mathbb{E}\Big[\sum_{i=0}^{\tau-1} d_i \,\Big|\, s, z\Big] \;=\; \sum_{i=0}^{\infty} \mathbb{E}\big[d_i \, \mathbf{1}\{i < \tau\} \mid s, z\big] \;=\; \sum_{i=0}^{\infty} \mathbb{E}\big[\, \mathbb{E}[d_i \mid \mathcal{F}_i] \, \mathbf{1}\{i < \tau\} \mid s, z\big],$$

where we used the tower property and the fact that $\mathbf{1}\{i < \tau\}$ is $\mathcal{F}_i$-measurable (since $\tau$ is a stopping time with respect to $\{\mathcal{F}_i\}$). Applying the bound on $\mathbb{E}[d_i \mid \mathcal{F}_i]$ gives

$$\mathbb{E}\Big[\sum_{i=0}^{\tau-1} d_i \,\Big|\, s, z\Big] \;\leq\; \sum_{i=0}^{\infty} \mathbb{E}\big[(\kappa + \xi)\,\mathbf{1}\{i < \tau\} \mid s, z\big] \;=\; (\kappa + \xi)\sum_{i=0}^{\infty} \Pr(i < \tau \mid s, z).$$

Finally, $\sum_{i=0}^{\infty} \mathbf{1}\{i < \tau\} = \tau$ almost surely, so by taking expectations we obtain $\sum_i \Pr(i < \tau \mid s, z) = \mathbb{E}[\tau \mid s, z]$. This proves the right inequality.

To prove equation 7, write $a h_\theta(s)$, $b g_\theta(h_\theta(s), z)$, and $c h_\theta(s')$. Since $d$ is a metric,

$$\big| d(a, b) - d(a, c) \big| \;\leq\; d(b, c).$$

Taking conditional expectation and using $|\,\mathbb{E}[X] - \mathbb{E}[Y]\,| \leq \mathbb{E}[\,|X - Y|\,]$,

$$\big| \mathbb{E}[d(a, b) \mid s, z] - \mathbb{E}[d(a, c) \mid s, z] \big| \;\leq\; \mathbb{E}[\,d(b, c) \mid s, z\,] \;\leq\; \bar{\varepsilon},$$

by Assumption B.1. This yields equation 7. $\qquad\square$

### B.4 Duration Lower Bound from the Learned Margin

We now show that combining the **High-Level Under-Displacement (Push)** term with the bound from Lemma B.6 establishes the learned margin $m_\phi(z)$ as a lower bound for the expected duration. This *derives* the geometry-duration coupling: the objective forces the latent representation to satisfy this property.

**Proposition B.7** (Expected Duration Lower Bound). *Suppose the high-level under-displacement hinge is satisfied up to an expected slack $\rho \geq 0$, i.e.,*

$$m_\phi(z) \;\leq\; \mathbb{E}\big[d_\theta(s, z) \mid s, z\big] \;+\; \rho.$$

*Under Assumptions B.1 and B.2, for any $(s, z)$,*

$$\boxed{\;\mathbb{E}[\tau \mid s, z] \;\geq\; \frac{m_\phi(z) - \rho - \bar{\varepsilon}}{\kappa + \xi}\;} \tag{8}$$

*(where the bound is informative when $m_\phi(z) > \rho + \bar{\varepsilon}$).*

*Proof.* From the hinge slack (consequence of minimizing the Push loss),

$$m_\phi(z) - \rho \;\leq\; \mathbb{E}\big[d_\theta(s, z) \mid s, z\big].$$

By equation 7 (triangle inequality on modeling error),

$$\mathbb{E}\big[d_\theta(s, z) \mid s, z\big] \;\leq\; \mathbb{E}\big[d(h_\theta(s), h_\theta(s')) \mid s, z\big] \;+\; \bar{\varepsilon}.$$

Combining the previous two displays yields

$$m_\phi(z) - \rho - \bar{\varepsilon} \;\leq\; \mathbb{E}\big[d(h_\theta(s), h_\theta(s')) \mid s, z\big].$$

Finally, apply the right inequality of equation 6 (consequence of the Pull/Cap loss):

$$\mathbb{E}\big[d(h_\theta(s), h_\theta(s')) \mid s, z\big] \;\leq\; \mathbb{E}[\tau \mid s, z]\,(\kappa + \xi),$$

and rearrange to obtain equation 8. $\qquad\square$

**Interpretation.** Proposition B.7 demonstrates that optimizing the unified objective forces the learned margin $m_\phi(z)$ to serve as a proxy for the expected duration $\mathbb{E}[\tau]$. Specifically, the margin scales with duration: larger margins are only geometrically permissible if the underlying transition takes longer (or moves faster, which is penalized by the Cap loss).

## B.5 ORDER CONSISTENCY VIA PAIRWISE HINGE

Finally, we show that the **Budget-Aware Order Consistency** term refines this coupling by ensuring the model's internal scorer $f_{\theta,\phi}$ correctly ranks subgoals by their effective duration, even if the bounds in Proposition B.7 are loose.

**Theorem B.8** (Budget-Aware Order Consistency). *Let $f_{\theta,\phi}$ be a population minimizer of the pairwise hinge term in Eq. (5), i.e., it minimizes*

$$\mathcal{R}_{\text{pair}}(f) = \mathbb{E}\Big[\max\{0,\, 1 - Y\left(f(s, z_i) - f(s, z_j)\right)\}\Big],$$

*where the expectation is over the joint distribution of $(s; z_i, b_i, z_j, b_j, Y)$. Under Assumption B.3, for almost every informative pair $(s; z_i, z_j)$,*

$$\text{sign}\big(f_{\theta,\phi}(s, z_i) - f_{\theta,\phi}(s, z_j)\big) = \text{sign}\big(\bar{\mu}(s, z_i) - \bar{\mu}(s, z_j)\big).$$

*Proof.* Fix an informative pair $(s; z_i, z_j)$ and write $\eta \Pr(Y = 1 \mid s; z_i, z_j) \in (0, 1) \setminus \{\frac{1}{2}\}$. Consider the conditional risk as a function of the scalar margin $u f(s, z_i) - f(s, z_j)$:

$$\varphi(u; \eta) = \eta\,(1 - u)_+ + (1 - \eta)\,(1 + u)_+,$$

where $(\cdot)_+ = \max\{0, \cdot\}$. The function $\varphi(\cdot; \eta)$ is convex and piecewise linear. We analyze its derivative on the three regions:

- For $u \leq -1$: $(1 - u)_+ = 1 - u$ and $(1 + u)_+ = 0$, so $\varphi(u; \eta) = \eta(1 - u)$ with derivative $\varphi'(u; \eta) = -\eta < 0$.

- For $-1 < u < 1$: $(1 - u)_+ = 1 - u$ and $(1 + u)_+ = 1 + u$, so $\varphi(u; \eta) = \eta(1 - u) + (1 - \eta)(1 + u) = 1 + (1 - 2\eta)u$ with derivative $\varphi'(u; \eta) = 1 - 2\eta$.

- For $u \geq 1$: $(1 - u)_+ = 0$ and $(1 + u)_+ = 1 + u$, so $\varphi(u; \eta) = (1 - \eta)(1 + u)$ with derivative $\varphi'(u; \eta) = 1 - \eta > 0$.

If $\eta > \frac{1}{2}$, then $1 - 2\eta < 0$; consequently, $\varphi'(\cdot; \eta)$ is negative on $(-1, 1)$, negative on $(-\infty, -1]$ (equal to $-\eta$), and positive on $[1, \infty)$ (equal to $1 - \eta$). Therefore the unique minimizer of $\varphi(\cdot; \eta)$ occurs at $u^\star = 1$ and, in particular, $\text{sign}(u^\star) = +1$. If $\eta < \frac{1}{2}$, a symmetric argument shows the unique minimizer is $u^\star = -1$ with $\text{sign}(u^\star) = -1$. If $\eta = \frac{1}{2}$, the set of minimizers is the whole interval $[-1, 1]$ and $\text{sign}(u^\star)$ may be $-1, 0$, or $+1$.

At the population level, minimizing $\mathcal{R}_{\text{pair}}(f)$ pointwise in $(s; z_i, z_j)$ yields the conditional minimizer $u^\star(s; z_i, z_j)$ with sign $\text{sign}(2\eta - 1)$. By Assumption B.3, $\eta > \frac{1}{2}$ iff $\bar{\mu}(s, z_i) > \bar{\mu}(s, z_j)$, and $\eta < \frac{1}{2}$ iff $\bar{\mu}(s, z_i) < \bar{\mu}(s, z_j)$. Thus the sign of the optimal margin $f(s, z_i) - f(s, z_j)$ coincides almost surely with the sign of $\bar{\mu}(s, z_i) - \bar{\mu}(s, z_j)$. $\qquad\square$

**A simple regret bound from hinge risk.** Let $z^\star(s) \in \arg\max_z V(s, z)$ and let $z_f(s) \in \arg\max_z f_{\theta,\phi}(s, z)$ be the planner's choice. Define the pairwise misordering event $\mathcal{E}(s; z^\star, z_f)\{(f(s, z^\star) - f(s, z_f))(\bar{\mu}(s, z^\star) - \bar{\mu}(s, z_f)) \leq 0\}$. Since $\mathbf{1}_{\{ab \leq 0\}} \leq \max\{0, 1 - \text{sign}(a)\text{sign}(b)\} \leq \max\{0, 1 - Y(f(s, z^\star) - f(s, z_f))\}$ with $Y = \text{sign}(\bar{\mu}(s, z^\star) - \bar{\mu}(s, z_f))$, we have $\mathbb{P}(\mathcal{E}) \leq \mathcal{R}_{\text{pair}}(f)$. Moreover,

$$V(s, z^\star) - V(s, z_f) \leq L_V\,|\bar{\mu}(s, z^\star) - \bar{\mu}(s, z_f)| \leq L_V\,\Delta_{\bar{\mu}}(s)\,\mathbf{1}_{\mathcal{E}(s; z^\star, z_f)},$$

where $\Delta_{\bar{\mu}}(s) = \max_{z,z'} |\bar{\mu}(s, z) - \bar{\mu}(s, z')|$. Taking expectations yields

$$\mathbb{E}\big[V(s, z^\star) - V(s, z_f)\big] \leq L_V\,\mathbb{E}[\Delta_{\bar{\mu}}(s)]\,\mathcal{R}_{\text{pair}}(f). \tag{9}$$

This complements Theorem B.8 by linking the surrogate risk directly to control performance.

## B.6 PLANNING REGRET FOR DEPTH-$H$ MCTS

We now quantify decision quality when using the learned model in a depth-$H$ search with $N$ simulations.

**Theorem B.9** (Planning Regret Bound). *Consider MCTS of depth $H$ over high-level actions, using $N$ simulations per decision. Suppose the leaf value estimator $\widehat{V}$ satisfies $\|\widehat{V} - V\|_\infty \leq \varepsilon_V$. Let $\varepsilon_T L_S \bar{\varepsilon}$ denote the per-macro-step value distortion induced by the model (Assumptions B.1 and B.4(iv)). Then, for any $\delta \in (0, 1)$, with probability at least $1 - \delta$,*

$$\text{Regret } \mathbb{E}\big[V(s, z^\star) - V(s, \hat{z})\big] \ \leq \ \tilde{\mathcal{O}}\left(\varepsilon_V \ + \ H\,\varepsilon_T \ + \ \sqrt{\frac{\log(|\mathcal{A}_{HL}|/\delta)}{N}}\right), \qquad (10)$$

*where $\hat{z}$ is the action selected by MCTS, $z^\star$ is the optimal action, $|\mathcal{A}_{HL}|$ is the cardinality of the high-level action set, and $\tilde{\mathcal{O}}$ hides absolute constants and logarithmic factors.*

*Proof.* We decompose the regret into three terms via a standard perturbation argument. Let $Q^\star(s, z)$ denote the true depth-$H$ return for selecting $z$ at the root and acting optimally thereafter in the true environment, and let $Q^{\mathrm{mod}}(s, z)$ denote the depth-$H$ return under the *modeled* dynamics that advance the latent state deterministically to $g_\theta(h_\theta(s), z)$ at each macro-step, but with the same reward function and discounting.[2] Let $\widehat{Q}(s, z)$ be the empirical MCTS estimate with leaf evaluation $\widehat{V}$ at depth $H$. We then write, for any $z$,

$$Q^\star(s, z) - \widehat{Q}(s, z) \ = \ \underbrace{Q^\star(s, z) - Q^{\mathrm{mod}}(s, z)}_{\text{model bias}} \ + \ \underbrace{Q^{\mathrm{mod}}(s, z) - \mathbb{E}[\widehat{Q}(s, z)]}_{\text{finite-horizon value bias}} \ + \ \underbrace{\mathbb{E}[\widehat{Q}(s, z)] - \widehat{Q}(s, z)}_{\text{Monte Carlo error}}.$$

Taking the supremum over $z$ controls the suboptimality of the $\arg\max_z \widehat{Q}(s, z)$ choice.

*(i) Model bias.)* By Assumptions B.1 and B.4(iv) and the tower property, at any node the next-state latent under the model, $g_\theta(h_\theta(s), z)$, differs in expectation from the true next-state latent $h_\theta(s')$ by at most $\bar{\varepsilon}$ in chordal distance. The Lipschitz property of $V$ in the latent geometry implies that the induced value distortion at that node is at most $L_S \bar{\varepsilon} = \varepsilon_T$. Unrolling over depth $H$ adds at most $H\,\varepsilon_T$ bias (subadditivity of absolute deviations), so

$$\sup_z \big| Q^\star(s, z) - Q^{\mathrm{mod}}(s, z) \big| \ \leq \ H\,\varepsilon_T.$$

*(ii) Finite-horizon value bias.)* Replacing the true value at the leaves with the approximate $\widehat{V}$ induces at most $\varepsilon_V$ error uniformly:

$$\sup_z \big| Q^{\mathrm{mod}}(s, z) - \mathbb{E}[\widehat{Q}(s, z)] \big| \ \leq \ \varepsilon_V,$$

since the only difference between these quantities is the terminal evaluation at depth $H$.

*(iii) Monte Carlo error.)* For UCT-style MCTS with bounded returns (Assumption B.4(i)), the empirical estimates concentrate uniformly over actions. A standard application of Hoeffding/Azuma inequalities (together with a union bound over $|\mathcal{A}_{HL}|$ root actions) gives

$$\Pr\left(\sup_z \big| \mathbb{E}[\widehat{Q}(s, z)] - \widehat{Q}(s, z) \big| \ > \ c\sqrt{\frac{\log(|\mathcal{A}_{HL}|/\delta)}{N}}\right) \ \leq \ \delta,$$

for an absolute constant $c > 0$ (the constant depends on the exploration bonus choice but not on $N$). See, e.g., standard UCT analyses.

Combining (i)–(iii) and using the fact that for any $\arg\max$ estimator $\hat{z}$,

$$Q^\star(s, z^\star) - Q^\star(s, \hat{z}) \ \leq \ \sup_z \big| Q^\star(s, z) - \widehat{Q}(s, z) \big| \ + \ \sup_z \big| \widehat{Q}(s, z) - Q^\star(s, z) \big|,$$

yields the stated bound equation 10, absorbing constants and polylog factors in $\tilde{\mathcal{O}}(\cdot)$. □

**Summary of Theoretical Results.**

1. **Lemma B.6** establishes that minimizing the *Low-Level Cap (Pull)* loss ensures the expected latent displacement is upper-bounded by the expected duration (scaled by $\kappa$).

---

[2] If rollouts accrue per-step rewards, these are evaluated along the corresponding (true vs. modeled) trajectories; Lipschitz smoothness in Assumption B.4(iv) transfers the transition discrepancy to return discrepancy.

2. **Proposition B.7** combines this with the *High-Level Margin (Push)* loss to prove that the learned margin $m_\phi(z)$ becomes a lower bound on the subgoal's expected duration. This *derives* the geometry-duration correlation rather than assuming it.

3. **Theorem B.8** shows that the *Order Consistency* loss ensures the model ranks subgoals correctly by duration.

4. **Theorem B.9** provides the final planning regret bound, decomposing error into value approximation, model bias, and search variance.

## C   BACKGROUND ON MUZERO-STYLE PLANNING IN GOALZERO

In this section, we detail a single simulation step of the high-level MCTS planner used in GoalZero. Unlike standard MuZero which operates on primitive actions, our planner searches over the space of abstract subgoals $\mathcal{Z}$. The search proceeds in a learned latent space defined by the representation network $h_\theta$, dynamics network $g_\theta$, and prediction network $f_\theta$.

Each MCTS simulation consists of three phases:

**1. Selection**   The search starts at the current root node $s^0 = h_\theta(s_{\text{env}})$. The algorithm traverses the tree by selecting the action (subgoal) $z^k$ that maximizes the Upper Confidence Bound (UCB) score:

$$z^k = \arg\max_{z \in \mathcal{Z}} \left[ Q(s^k, z) + P(s^k, z) \cdot \frac{\sqrt{\sum_b N(s^k, b)}}{1 + N(s^k, z)} \cdot \left( c_1 + \log\left( \frac{\sum_b N(s^k, b) + c_2}{c_2} \right) \right) \right],$$

where $Q(s, z)$ is the mean estimated value of taking subgoal $z$ from state $s$, $P(s, z)$ is the prior probability from the policy head $f_\theta^\pi$, and $N(s, z)$ is the visit count. This selection repeats until a leaf node $s^l$ is reached.

**2. Expansion and Recurrent Inference**   Upon reaching a leaf node $s^l$ that has not yet been expanded, we use the dynamics function $g_\theta$ to simulate the SMDP transition:

$$s^{l+1}, \hat{R}^l = g_\theta(s^l, z^l).$$

Here, $s^{l+1}$ is the predicted next latent state after the subgoal execution, and $\hat{R}^l$ is the predicted cumulative discounted reward accumulated during the variable duration of the subgoal. We then evaluate $s^{l+1}$ using the prediction function:

$$\mathbf{p}^{l+1}, v^{l+1} = f_\theta(s^{l+1}),$$

where $\mathbf{p}^{l+1}$ initializes the priors for the new node and $v^{l+1}$ estimates the long-term value from that state onwards.

**3. Backup**   The estimated value $v^{l+1}$ and the rewards $\hat{R}$ along the path are propagated backward to update the statistics of all nodes traversed during the selection phase. For a node at depth $k < l$, the return is computed as:

$$G^k = \sum_{\tau=0}^{l-1-k} \gamma^\tau \hat{R}^{k+\tau} + \gamma^{l-k} v^{l+1}.$$

The Q-values are updated as a moving average: $Q(s^k, z^k) \leftarrow \frac{N(s^k, z^k) \cdot Q(s^k, z^k) + G^k}{N(s^k, z^k) + 1}$, and visit counts are incremented.

After $N_{\text{sim}}$ simulations, the real high-level action is selected by sampling from the visit count distribution at the root, $\pi_{\text{MCTS}}(z) \propto N(s^0, z)^{1/T_{\text{temp}}}$.

## D   ALGORITHM

## E   ENVIRONMENTS

We evaluate on two SSCO-style benchmarks following Feng et al. (2025): Adaptive Influence Maximization (AIM) and Stochastic Orienteering (SOP). In both settings, the higher layer allocates a *budget* at each decision point, and the lower layer selects items (nodes or cities) conditioned on that budget.

---

**Algorithm 1** GoalZero Training Loop

---

1: Initialize networks: $h_\theta, f_\theta, g_\theta, m_\phi, b_\psi$, and LL policy $\pi^L$.
2: Initialize replay buffers $\mathcal{D}_{\text{HL}}$ (for games) and $\mathcal{D}_{\text{LL}}$ (for primitive transitions).
3: **for** each training epoch **do**
4:  // — **Phase 1: Self-Play Experience Collection** —
5:  Run one full episode in the environment to generate a game history.
6:  **for** each HL step $k$ in the episode **do**
7:    Get current state $s_k$.
8:    Run MCTS using $h_\theta, f_\theta, g_\theta$ to get MCTS policy $\pi_k$ and value $v_k$ targets.
9:    Sample subgoal $z_k \sim \pi_k$.
10:    Sample budget $b_k \sim b_\psi(\cdot|h_\theta(s_k), z_k)$.
11:    Execute LL policy $\pi^L(\cdot|s_k, z_k)$ for up to $b_k$ steps.
12:    Collect cumulative reward $R_k$ and resulting state $s_{k+1}$.
13:    Store HL transition $(s_k, z_k, R_k, s_{k+1})$ and MCTS targets $(\pi_k, v_k)$ in the game history.
14:    Store all collected primitive transitions $(s_t, z_k, a_t, r_t, s_{t+1})$ in $\mathcal{D}_{\text{LL}}$.
15:  **end for**
16:  Add the completed game history to $\mathcal{D}_{\text{HL}}$.
17:  // — **Phase 2: High-Level World Model Update** —
18:  **if** $|\mathcal{D}_{\text{HL}}|$ and $|\mathcal{D}_{\text{LL}}|$ are large enough **then**
19:    Sample a batch of game trajectories from $\mathcal{D}_{\text{HL}}$.
20:    Sample a batch of primitive transitions from $\mathcal{D}_{\text{LL}}$.
21:    Compute standard MuZero losses ($L_{\text{policy}}, L_{\text{value}}, L_{\text{reward}}$) on the HL batch.
22:    Compute actor-critic losses for the budget head $b_\psi$.
23:    Compute loss $\mathcal{L}_{\text{MTS}}$ from Eq. (5).
24:    Update HL networks ($h_\theta, f_\theta, g_\theta, m_\phi, b_\psi$) by minimizing all HL losses.
25:  **end if**
26:  // — **Phase 3: Low-Level Policy Update** —
27:  **if** $|\mathcal{D}_{\text{LL}}|$ is large enough **then**
28:    Sample a batch of primitive transitions from $\mathcal{D}_{\text{LL}}$.
29:    Update LL policy $\pi^L$ by minimizing the DQN loss $\mathcal{L}_{LL}$.
30:  **end if**
31: **end for**

---

**Adaptive Influence Maximization (AIM).** AIM extends classical IM to progressive, uncertain settings under the Independent Cascade (IC) model. The environment is a directed graph $G = (V, E)$. Each node is inactive/active/removed. When a node becomes active (seeded or activated), it attempts to activate each inactive neighbor on the next day with probability $p$, after which it cannot activate further. Following Feng et al. (2025), $p$ is set inversely to indegree, and graphs are randomly generated. A total seed budget and a time horizon are given; before each day, the agent selects seed nodes subject to the remaining budget. The objective is to maximize the number of influenced nodes within the horizon.

At the higher layer, the state comprises node states (inactive/active/removed) and global information (e.g., remaining budget); the option is the budget allocated for the current day; the reward is the increase in influenced nodes. The lower layer selects nodes given the allocated budget; state transitions apply seed selections and one-step IC activations.

**Stochastic Orienteering (SOP).** SOP is a travel-planning variant with a set of cities (coordinates sampled i.i.d.), per-city profits that evolve over time, and a daily travel-distance limit with penalties for exceedance. The agent must choose which cities to visit over a fixed number of days, starting and ending at a designated city, to maximize total profit. A submodular aggregation is used for the one-day profit over selected cities; profits reset to zero upon visit and otherwise evolve stochastically as in Feng et al. (2025).

At the higher layer, the state includes city coordinates, current profits, the current city, and the start city; the option is the budget allocated to the current day; the reward is the combined (submodular) profit from cities selected that day. The lower layer selects cities up to the allocated budget; the transition updates profits and current location and applies the travel-penalty rule when the daily limit is exceeded.

## F NETWORK ARCHITECTURE DETAILS

Our framework is composed of two main hierarchical levels: the high-level planner and the low-level policy. Their architectures are detailed below.[3]

**High-Level Networks.** The high-level planner consists of three core components operating in the latent space:

**Representation ($h_\theta$)** A two-layer message-passing Graph Neural Network (GNN) using linear aggregation with the graph adjacency matrix. The GNN is followed by ReLU activations, graph-level mean pooling, and a final linear projection to the $d$-dimensional latent space. The output is $\ell_2$-normalized on every forward pass to produce $\bar{h}_\theta(s)$.

**Dynamics ($g_\theta$)** A Multi-Layer Perceptron (MLP) that takes the concatenation of the normalized latent state $\bar{h}_\theta(s)$ and the subgoal embedding $z$ as input. It outputs the subsequent latent state, which is also $\ell_2$-normalized. The normalized input and output ensure that both $\bar{h}$ and $\bar{g}$ lie on the unit hypersphere $\mathbb{S}^{d-1}$.

**Prediction ($f_\theta$)** This component consists of four separate heads that operate on the latent state:

- **Policy Head:** An MLP predicting a categorical distribution over the discrete set of subgoals $\mathcal{Z}$.
- **Value Head:** An MLP predicting the scalar expected return from the current state.
- **Reward Head:** An MLP that takes the concatenation of the latent state and subgoal, $\mathrm{concat}(\bar{h}, z)$, to predict the cumulative reward over the subgoal's duration as a categorical support.
- **Budget Head:** An MLP, also conditioned on $\mathrm{concat}(\bar{h}, z)$, that outputs a discrete distribution over possible budget allocations. This head is trained via an actor-critic objective.

**Low-Level Policy ($\pi^L$).** The low-level policy is implemented as a subgoal-conditioned GNN-based Q-network. It processes the graph state and computes Q-values for each primitive action (e.g., selecting a node). The GNN's node representations are concatenated with the current high-level subgoal embedding $z$ to make the policy goal-aware. The network employs illegal-action masking to ensure that only valid actions are selected during execution.

---

[3]Code is available at https://anonymous.4open.science/r/GoalZero-HRL/.

Table 4: Key training hyperparameters (shared across AIM and SOP unless noted).

| Component | Hyperparameter | Symbol | Value |
|---|---|---|---|
| Latent Space | Latent dimension | $d$ | 128 |
| MuZero-style | Learned subgoals | $|\mathcal{Z}|$ | 32 |
| HL Planner | Unroll steps for training | $K$ | 5 |
| | LL per-step cap (unit sphere) | $\kappa$ | 0.10 |
| MTS–SMDP World | Margin regularizer (L2) | $\lambda_m$ | $1 \times 10^{-4}$ |
| Model (Unified Loss) | Loss weight (cap / Pull) | $w_{\text{cap}}$ | 0.10 |
| | Loss weight (push / margin) | $w_{\text{push}}$ | 0.10 |
| | Loss weight (order / pairwise) | $w_{\text{order}}$ | 0.10 |
| Low-Level (LL) | $\varepsilon$-greedy schedule | $\varepsilon$ | $0.90 \rightarrow 0.05$ (exp., decay 0.995) |
| Controller | Discount factor | $\gamma$ | 0.997 |
| | Optimizer (all nets) | – | Adam |
| Optimization / | HL learning rate (repr/dyn/pred/margin/budget) | $\eta_{\text{HL}}$ | $1 \times 10^{-3}$ |
| Replay | LL learning rate | $\eta_{\text{LL}}$ | *see Tab. 5* |
| | Weight decay (all) | – | $1 \times 10^{-5}$ |
| | Gradient clipping | – | global norm 5.0 |
| | Batch size (HL / LL) | – | 8 (games/unrolls) / 8 (transitions) |
| | HL replay buffer size (games) | – | 1,000 |
| Buffers / | LL replay buffer size (steps) | – | 10,000 |
| Data | Seeds per configuration | – | 10 |
| | Evaluation metric | – | mean $\pm$ s.e.m. |

Table 5: Domain-specific overrides. Values omitted here inherit from Table 4.

| Domain | Hyperparameter | Symbol | AIM | SOP |
|---|---|---|---|---|
| HL Planner | MCTS simulations / HL decision | $N_{\text{sim}}$ | 150 | 250 |
| HL Planner | PUCT exploration constant (init) | $c_{\text{init}}$ | 2.5 | 1.5 |
| Exploration | MCTS root noise (training) | – | Dirichlet $\alpha = 0.30$, mix $\epsilon = 0.30$ | Dirichlet $\alpha = 0.20$, mix $\epsilon = 0.20$ |
| Training | Search temperature (training) | $\tau_{\text{search}}$ | 1.0 | anneal $1.5 \rightarrow 0.5$ |
| Optimization | LL learning rate | $\eta_{\text{LL}}$ | $1 \times 10^{-3}$ | $5 \times 10^{-4}$ |
| Budget AC | Entropy bonus | $\beta$ | 0.01 | 0.02 (annealed) |

## G  TRAINING DETAILS FOR THE BUDGET ACTOR–CRITIC

We parameterize $b_\psi(b \mid s, z) = \text{softmax}(W_2\, \sigma(W_1[\bar{h}_\theta(s); z]))$ over budgets $b \in \{0, \ldots, B_{\max}\}$. At each HL decision the planner outputs $(s_t, z_t)$; we sample $b_t \sim b_\psi(\cdot | s_t, z_t)$, roll out the LL controller for up to $b_t$ micro-steps, and back up a return $G_t$ from the search. The critic $V_\eta(s, z)$ minimizes $\mathbb{E}[(G_t - V_\eta(s_t, z_t))^2]$. The actor maximizes $\mathbb{E}[\log b_\psi(b_t | s_t, z_t)\,(G_t - V_\eta(s_t, z_t)) + \beta\,\mathcal{H}]$. We interleave actor–critic updates with representation/dynamics losses (including Eq. (5)). For stability we use target networks for $V_\eta$, and entropy $\beta \in [0.01, 0.1]$.

For stability, we use a brief warm-up of three epochs before enabling the learned budget head $b_\psi(\cdot \mid s, z)$. During this warm-up the high-level planner still selects subgoals via MCTS and *all* high-level networks ($h_\theta, g_\theta, f_\theta$) and the low-level controller continue training from self-play, but the budget is sampled from a simple heuristic (average allocation). This delays credit assignment to the budget actor for just a few epochs, which empirically reduces early high-variance updates and avoids coupling instabilities between subgoal search and budget allocation, while *preserving* end-to-end gradients and on-policy targets for the world model from the outset. Conceptually, this is a lightweight alternative to the *wake–sleep* schedule in WS-option (Feng et al., 2025), where the "sleep" stage freezes the high-level budget policy and uses an average allocation so the low level can approach convergence before joint training resumes (and the high level is trained with MC targets to mitigate error propagation). In contrast, our three-epoch warm-up does *not* freeze any heads and does not pause high-level learning; it simply postpones learning of $b_\psi$ to dampen early interference, after which actor–critic training of the budget head is enabled and proceeds jointly with the rest of the system.

## H  ADDITIONAL ABLATIONS ON MTS LOSS TERMS

We isolate the three components of the unified MTS–SMDP objective in Eq. (5) and report average reward (mean $\pm$ s.e.m.) with $N{=}500$, $K{=}70$, and 10 seeds: (i) **Pull** = LL per-step cap (local geometric smoothness), (ii) **Push** = HL under-displacement margin (prevents vanishing macro steps),

(iii) **Order** = budget-aware pairwise hinge (rank-calibrates durations). Each ablated variant is trained from scratch with the same protocol as the main experiments.

Table 6: AIM and SOP ($N{=}500, K{=}70$): Per-term loss ablation. Average reward $\pm$ s.e.m. (10 seeds).

| Variant | AIM: Avg. Reward | SOP: Avg. Reward |
|---|---|---|
| **Full (Ours)** | **324.15** $\pm$2.38 | **31.60** $\pm$1.94 |
| w/o **Pull** (LL cap) | 318.72 $\pm$3.20 | 24.66 $\pm$1.55 |
| w/o **Push** (margin) | 319.61 $\pm$2.95 | 25.14 $\pm$1.48 |
| w/o **Order** (pairwise hinge) | 320.17 $\pm$2.92 | 26.48 $\pm$1.52 |

As shown in Table 6, removing any single term reduces average reward, with the **LL cap (Pull)** being most critical (largest drop), followed by the **margin (Push)**, and then the **pairwise Order** term. This ordering is pronounced on SOP, where local geometric control and duration-aware macro steps are essential for routing under uncertainty. These trends align with our analysis: the cap establishes macro–micro coupling (Lem. B.6), the margin provides a duration-sensitive lower bound on macro steps (Prop. B.7), and the budget-aware pairwise hinge refines duration calibration (Thm. B.8).

## I   HYPER-PARAMETER SENSITIVITY

We investigate four knobs: number of learned subgoals $|\mathcal{Z}|$, MCTS simulations per HL decision $N_{\text{sim}}$, unroll steps $K$, and the LL per-step cap $\kappa$.

**Grids.**   $|\mathcal{Z}| \in \{16, 32, 64\}$,   $N_{\text{sim}}^{\text{AIM}} \in \{50, 100, 150, 200\}$,   $N_{\text{sim}}^{\text{SOP}} \in \{150, 200, 250, 300\}$, $K \in \{3, 5, 7\}$,   $\kappa \in \{0.05, 0.10, 0.15\}$. For each setting we keep total wall-clock within a budget by proportionally adjusting batch sizes. Results are reported in Table 7, Tables 8–9, Table 10, and Table 11.

Table 7: Sensitivity to number of learned subgoals $|\mathcal{Z}|$ (10 seeds). Means $\pm$ s.e.m. are reported.

| $|\mathcal{Z}|$ | 16 | 32 | 64 |
|---|---|---|---|
| AIM | 320.85 $\pm$3.25 | 324.15 $\pm$2.38 | 323.89 $\pm$2.56 |
| SOP | 29.13 $\pm$1.16 | 31.60 $\pm$1.94 | 31.52 $\pm$1.39 |

Table 8: AIM: Sensitivity to MCTS simulations $N_{\text{sim}}$ (10 seeds). Means $\pm$ s.e.m. are reported.

| $N_{\text{sim}}$ | 50 | 100 | 150 | 200 |
|---|---|---|---|---|
| AIM | 322.05 $\pm$3.88 | 323.92 $\pm$2.45 | 324.15 $\pm$2.38 | 324.03 $\pm$2.21 |

*Remark (cap sensitivity).* The LL cap sets the per-step latent-motion scale that enters the theoretical coupling; performance varies within $\sim$1 s.e.m. across the sweep, suggesting low sensitivity once the scale is reasonable.

## J   ZERO-SHOT GENERALIZATION TO LARGE GRAPHS

Tables 12 and 13 report large-graph generalization for agents trained once on $N{=}100$ graphs with $(T, K){=}(10, 60)$ and evaluated *zero-shot* on unseen graphs with $N \in \{1000, 1500, 2000, 2500\}$, using identical settings.

In AIM, GoalZero consistently outperforms all baselines, with the margin increasing as $N$ grows; degree/score heuristics remain competitive only at smaller $N$, while WS-option and Flat DQN degrade with action-space scale and propagation depth. In SOP, GoalZero is the only learning-based method that reliably surpasses the strong `Greedy` heuristic on the largest graphs, whereas WS-option and Flat DQN struggle to maintain globally coherent routes as instance size increases. We attribute this robustness to macro-level MuZero-style planning combined with a multi-timescale latent geometry that enforces local smoothness yet preserves semantically large subgoal displacements, enabling temporally calibrated, single-step HL evaluations that transfer without re-tuning.

Table 9: SOP: Sensitivity to MCTS simulations $N_{\text{sim}}$ (10 seeds). Means $\pm$ s.e.m. are reported.

| $N_{\text{sim}}$ | 150 | 200 | 250 | 300 |
|---|---|---|---|---|
| SOP | 30.91 $\pm$2.18 | 31.27 $\pm$1.25 | 31.60 $\pm$1.94 | 31.44 $\pm$1.36 |

Table 10: Sensitivity to unroll steps $K$ (10 seeds). Means $\pm$ s.e.m. are reported.

| $K$ | 3 | 5 | 7 |
|---|---|---|---|
| AIM | 322.93 $\pm$3.95 | 324.15 $\pm$2.38 | 324.02 $\pm$2.72 |
| SOP | 29.91 $\pm$2.21 | 31.60 $\pm$1.94 | 31.45 $\pm$2.28 |

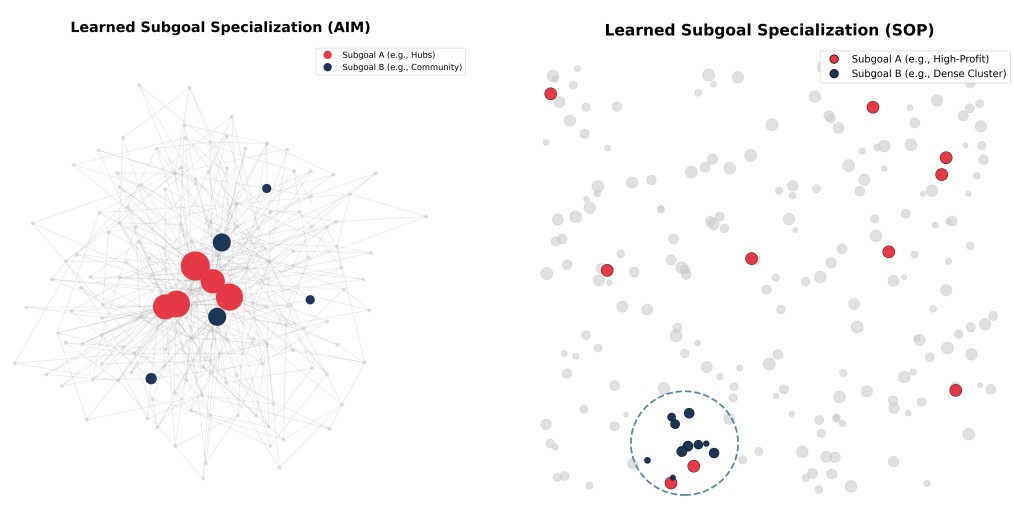

(a) AIM: Hub-Hunting vs. Community-Targeting

(b) SOP: High-Profit vs. Dense-Cluster

Figure 3: Visualization of learned subgoal specialization. In both domains, GoalZero learns distinct and interpretable strategies. (a) In AIM, one subgoal targets high-degree central hubs (red) while another focuses on a dense community (blue). (b) In SOP, one subgoal prioritizes scattered high-profit nodes (red), whereas another concentrates on a spatially dense neighborhood (blue; dashed circle), illustrating region-level specialization.

# K QUALITATIVE ANALYSIS OF SUBGOAL SPECIALIZATION

Beyond quantitative metrics, we seek to understand if the learned subgoals correspond to qualitatively distinct and strategically meaningful behaviors. To investigate this, we visualize the ideal actions a specialized low-level policy would take when conditioned on different subgoals in both the AIM and SOP domains, as shown in Figure 3. In both visualizations, the size of a node is proportional to its intrinsic value in the problem: its degree in the AIM task and its profit in the SOP task.

Figure 3a illustrates this for the AIM task on an exemplar graph. The visualization shows a clear strategic divergence: one subgoal (red) learns to target the central, high-degree hubs for broad, high-impact influence, which are visibly the largest nodes. In contrast, the other subgoal (blue) focuses on a dense local community of smaller nodes, representing a more focused, saturating strategy. Similarly, Figure 3b shows the emergent strategies for the SOP task. One subgoal (red) identifies high-profit targets scattered across the map, which are clearly depicted as larger nodes, while the second subgoal (blue) identifies a dense cluster of smaller, lower-profit nodes, prioritizing travel efficiency. Together, these visualizations demonstrate that GoalZero's abstract subgoals can correspond to concrete, interpretable, and strategically diverse plans.

Table 11: Sensitivity to the LL cap $\kappa$ (10 seeds). Means $\pm$ s.e.m. are reported.

| $\kappa$ | 0.05 | 0.10 | 0.15 |
|---|---|---|---|
| AIM: Avg. Reward | 324.01 $\pm$2.86 | 324.15 $\pm$2.38 | 323.94 $\pm$2.64 |
| SOP: Avg. Reward | 31.32 $\pm$1.82 | 31.60 $\pm$1.94 | 31.41 $\pm$1.85 |

Table 12: AIM: Generalization to larger graphs (trained on graph $N = 100$, $T = 10$, $K = 60$). All improvements of GoalZero over the best baseline are statistically significant (p-value $\leq 0.05$).

| Method | $N = 1000$ | $N = 1500$ | $N = 2000$ | $N = 2500$ |
|---|---|---|---|---|
| **GoalZero (Ours)** | **416.06** $\pm$**5.81** | **540.78** $\pm$**6.92** | **605.55** $\pm$**7.54** | **654.29** $\pm$**8.11** |
| WS-option | 370.15 $\pm$5.25 | 465.22 $\pm$6.13 | 510.98 $\pm$6.88 | 531.40 $\pm$7.21 |
| Flat DQN | 341.82 $\pm$5.40 | 431.65 $\pm$6.12 | 482.37 $\pm$6.78 | 501.93 $\pm$7.05 |
| average-degree | 366.36 $\pm$5.11 | 493.57 $\pm$6.45 | 549.02 $\pm$7.02 | 590.10 $\pm$7.63 |
| average-score | 387.80 $\pm$5.43 | 498.78 $\pm$6.51 | 572.56 $\pm$7.28 | 617.14 $\pm$7.95 |
| normal-degree | 166.01 $\pm$3.11 | 174.82 $\pm$3.28 | 180.55 $\pm$3.41 | 184.61 $\pm$3.52 |
| normal-score | 178.95 $\pm$3.35 | 189.42 $\pm$3.55 | 197.09 $\pm$3.71 | 201.95 $\pm$3.83 |
| static-degree | 355.20 $\pm$4.98 | 470.11 $\pm$6.21 | 521.73 $\pm$6.91 | 560.88 $\pm$7.44 |
| static-score | 371.44 $\pm$5.18 | 482.90 $\pm$6.33 | 550.16 $\pm$7.09 | 599.23 $\pm$7.77 |

Table 13: SOP: Generalization to larger graphs (trained on $N = 100$, $T = 10$, $K = 60$). All improvements of GoalZero over the best baseline are statistically significant (p-value $\leq 0.05$).

| Method | $N = 1000$ | $N = 1500$ | $N = 2000$ | $N = 2500$ |
|---|---|---|---|---|
| **GoalZero (Ours)** | **31.27** $\pm$**1.21** | **33.39** $\pm$**1.29** | **36.15** $\pm$**1.38** | **38.88** $\pm$**1.45** |
| WS-option | 18.63 $\pm$0.99 | 19.07 $\pm$1.03 | 20.52 $\pm$1.11 | 21.23 $\pm$1.15 |
| Flat DQN | 20.41 $\pm$1.06 | 22.18 $\pm$1.12 | 23.95 $\pm$1.18 | 25.72 $\pm$1.22 |
| Greedy | 23.12 $\pm$1.10 | 26.05 $\pm$1.19 | 27.37 $\pm$1.24 | 29.04 $\pm$1.30 |
| GA | 12.62 $\pm$0.85 | 10.96 $\pm$0.79 | 11.37 $\pm$0.81 | 11.77 $\pm$0.83 |

## L    VALIDATION OF THE MULTI-TIMESCALE WORLD MODEL VIA KENDALL'S TAU

We empirically investigate whether the Multi-Timescale SMDP (MTS) objective enables the agent to learn a temporally meaningful latent space, where the model's internal score for a subgoal, $\bar{f}(s, z)$, is monotonically related to the subgoal's true duration, $\hat{\tau}(s, z)$. To this end, we use Kendall's Rank Correlation Coefficient ($\tau$). A high positive correlation ($\tau \rightarrow 1$) indicates that the model has successfully learned the relative temporal ordering of its abstract actions, a crucial capability for effective long-term planning.

### L.1    ORDER-CONSISTENCY OF THE MTS OBJECTIVE

We analyze the quality of the learned temporal ordering for the two variants of our framework that incorporate the MTS objective. Table 14 presents the Kendall's Tau correlation for both the fixed-margin version (Variant D) and the full GoalZero model with a learnable margin (Variant E). The results provide clear, quantitative evidence that the MTS objective is highly effective. Both variants achieve a strong positive correlation, with Tau values consistently in the $0.69 \sim 0.76$ range. This indicates that the model has successfully learned a reliable sense of temporal order. Furthermore, the full GoalZero model (E) consistently achieves a higher correlation than the fixed-margin version (D), demonstrating the benefit of the learnable, subgoal-specific margin in further refining the temporal representation.

Table 14: Kendall's Tau correlation between the learned score $f(s, z)$ and empirical duration $\hat{\tau}(s, z)$ for the variants incorporating the MTS objective. Both achieve a strong positive correlation, directly validating the effectiveness of our proposed mechanism.

| Algorithm Variant | AIM Kendall's $\tau$ | SOP Kendall's $\tau$ |
|---|---|---|
| D: + MTS (Fixed Margin) | 0.71 $\pm$0.04 | 0.69 $\pm$0.05 |
| E: GoalZero (Full Model) | 0.74 $\pm$0.03 | 0.76 $\pm$0.04 |

Table 15: Calibration of value–geometry smoothness. Rank correlations between chordal distance $d(u, v)$ and $|\Delta V|$ with bootstrap 95% CIs. Each task uses $n=125$ points (5 seeds $\times$ 25 epochs), $T=10$, $K=60$. These empirical rank relations complement Assumption B.4 and the monotone-ordering guarantees (Lemma B.6, Proposition B.7, and Theorem B.8), which underpin the regret bound in Theorem B.9.

| Task | Kendall's $\tau_b$ [95% CI] | Spearman's $\rho$ [95% CI] |
|------|------------------------------|-----------------------------|
| AIM | **0.493** [0.401, 0.580] | **0.666** [0.536, 0.779] |
| SOP | **0.559** [0.475, 0.629] | **0.748** [0.634, 0.813] |

### L.2 CALCULATION METHODOLOGY

The reported Kendall's Tau values were generated following a procedure for each of the 10 random seeds per algorithm variant.

1. **Model Selection:** We take the final trained agent from each of the 10 independent runs.

2. **Data Sampling:** We perform rollouts on a held-out set of 50 test environment instances. During these rollouts, we sample 100 high-level states $s$ at random.

3. **Generating Paired Rankings:** For each sampled state $s$, we generate two ranked lists over the entire set of available subgoals $\mathcal{Z} = \{z_1, z_2, ..., z_{|\mathcal{Z}|}\}$:
   - **Learned Score Ranking:** We perform a forward pass for each subgoal to compute the list of learned scores $\{f(s, z_1), f(s, z_2), ..., f(s, z_{|\mathcal{Z}|})\}$.
   - **Empirical Duration Ranking:** To get a stable estimate of the ground-truth duration, we execute each subgoal $z_i$ from state $s$ five separate times, allowing the low-level policy to run until termination. We record the number of steps taken in each of the five runs and use the average duration to form the list $\{\hat{\tau}(s, z_1), \hat{\tau}(s, z_2), ..., \hat{\tau}(s, z_{|\mathcal{Z}|})\}$. This averaging mitigates the effects of stochasticity in both the environment and the low-level policy.

4. **Calculation and Aggregation:** For each of the 100 sampled states, we compute the Kendall's Tau correlation coefficient between the score ranking and the duration ranking. The final Tau value for a single seed is the average of these 100 individual Tau calculations.

5. **Final Reporting:** The values presented in the tables are the mean and standard error of the mean (s.e.m.) of the final Tau values from the 10 independent seeds.

This comprehensive process ensures that the reported correlation is a robust and reliable measure of the model's learned temporal awareness.

## M CALIBRATION OF VALUE–GEOMETRY SMOOTHNESS

We empirically examine the link between latent displacement and value variation posited by our smoothness assumptions. At each high-level decision point we form pairs $(u, v)$ with $u = h_\theta(s)$ and $v = g_\theta(h_\theta(s), z)$, and compute the chordal distance on the unit sphere, $d(u, v) = \|u - v\|_2$. We estimate $|\Delta V| \triangleq |V(u) - V(v)|$ using the same training targets as in the main method. Unless stated otherwise, results aggregate **5 seeds** over **25 epochs** per task (total $n = 125$ points) under the task settings $T=10$, $K=60$.

We report Kendall's $\tau_b$ and Spearman's $\rho$ between $d(u, v)$ and $|\Delta V|$, along with nonparametric bootstrap 95% confidence intervals. Both benchmarks show a clear positive monotonic association (Table 15); the corresponding scatter plots are in Fig. 4. The strong rank correlations imply that larger latent displacements are associated with larger absolute value changes, supporting the use of latent geometry as a proxy for temporal/strategic scale in SMDP planning.

## N ADDITIONAL RESULTS

We report results for two new baselines and one new task to further validate the efficacy of GoalZero.

**Baselines.**

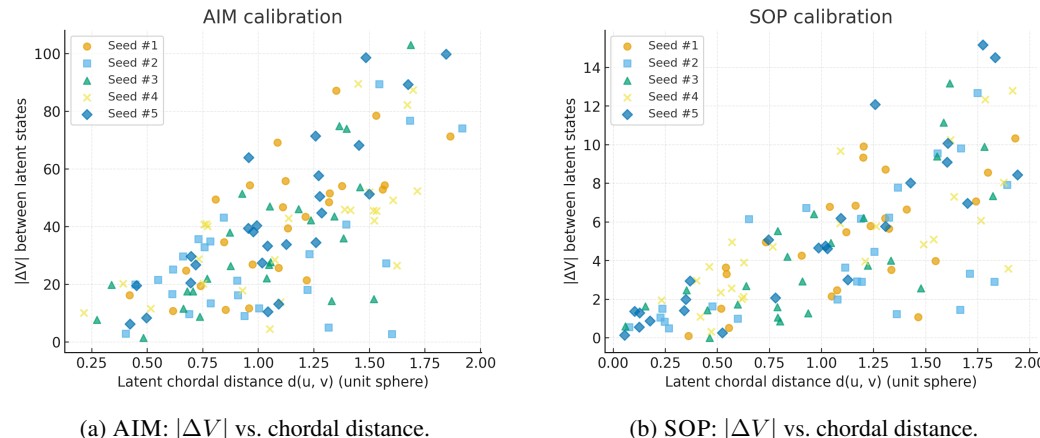

(a) AIM: $|\Delta V|$ vs. chordal distance.  (b) SOP: $|\Delta V|$ vs. chordal distance.

Figure 4: Calibration plots probing the value–geometry relationship. Points aggregate 5 seeds $\times$ 25 epochs per task.

- **Director** (Hafner, 2022): A goal-conditioned hierarchical model-based RL agent. Since Director uses a fixed high-level stride $H$, we adapted it to SSCO by using an even budget allocation strategy (budget per stage is fixed).

- **Flat MuZero (Schrittwieser et al., 2020)**: A standard model-based RL agent without hierarchy, operating directly on primitive actions using the same GNN encoder as GoalZero.

**Additional Task: Power-2500.**  We evaluate all learning-based methods on the **Power-2500** task, which is based on the large-scale influence maximization instance from Feng et al. (2025) with graph size $N = 2500$, but with a significantly higher budget ($K = 60$).

**Quantitative Results.**  Table 16 summarizes the performance. GoalZero consistently outperforms both Director and Flat MuZero. Director struggles due to its inability to dynamically allocate budgets, while Flat MuZero suffers from the large combinatorial action space.

Table 16: Comparison with additional model-based baselines on AIM, SOP, and the Power-2500 task. Mean $\pm$ s.e.m. over 10 seeds.

| Method | AIM ($N = 500, K = 70$) | SOP ($N = 500, K = 70$) | Power ($N = 2500, K = 60$) |
|---|---|---|---|
| **GoalZero** | **324.15** $\pm$2.38 | **31.60** $\pm$1.94 | **856.14** $\pm$10.20 |
| WS-Option | 301.53 $\pm$9.10 | 12.99 $\pm$1.25 | 676.86 $\pm$21.58 |
| Flat DQN | 243.46 $\pm$3.78 | 15.90 $\pm$0.67 | 464.78 $\pm$14.28 |
| Director | 306.53 $\pm$2.58 | 20.74 $\pm$0.93 | 725.61 $\pm$5.00 |
| Flat MuZero | 260.60 $\pm$8.41 | 17.75 $\pm$0.30 | 354.42 $\pm$4.32 |

**Additional Learning Curves.**  Figure 5 shows the learning curves for the $K = 70$ settings (moved from the main text) and the Power-2500 task.

## O  COMPUTATIONAL COST

We provide a detailed breakdown of GoalZero's computational cost and a comparison against baselines.

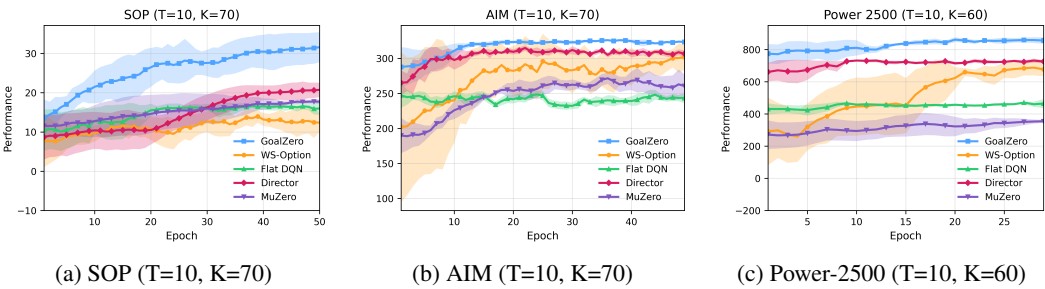

(a) SOP (T=10, K=70)  (b) AIM (T=10, K=70)  (c) Power-2500 (T=10, K=60)

Figure 5: Learning curves for the most challenging settings, including the Power-2500 task.

Table 17: Setup for GoalZero profiling.

| Item | Value |
|---|---|
| GPU | NVIDIA Tesla V100 (32 GB HBM2) |
| CUDA / cuDNN | 11.7 / 8.5 |
| PyTorch | 2.0.1 |
| Python | 3.8.19 |
| OS | Linux (kernel 4.15.0-135-generic) |
| Mixed precision (AMP) | disabled |
| Task | AIM with $N = 500, T = 10, K = 70$ |
| Subgoals / MCTS simulations | 32 / 150 |

## O.1 GOALZERO DETAILED METRICS

Notes: decision latency includes MCTS, subgoal selection, budget selection, and low-level action selection. Episode latency sums per-decision latency over $T$ decisions. CUDA synchronization is applied around timed regions.

## O.2 COMPARISON WITH BASELINES

We compare the computational cost of GoalZero against baselines in Table 19. Measurements were taken on the same hardware setup described above.

**Analysis:** GoalZero incurs higher latency than model-free methods (DQN, WS-Option) due to the MCTS planning overhead. Crucially, GoalZero is significantly more sample-efficient (as shown in Fig. 2), achieving higher rewards with fewer environment interactions, which justifies the additional compute during planning.

Table 18: GoalZero Results.

| Metric | Value |
| --- | --- |
| Training time / epoch | 62.26 s |
| Decision latency (median / p95) | 441.0 ms / 519.5 ms |
| Episode latency (mean; $T{=}10$) | 4,288.4 ms |
| Episodes / epoch | 16 |
| Env micro-steps (true frames) | 55,154 |
| Peak GPU memory (training / inference) | 0.28 GB / 0.02 GB |

Table 19: Wall-clock time comparison (AIM, $N = 500$, $T = 10$, $K = 70$).

| Method | Decision Latency (ms) | Training Time / Epoch (s) |
| --- | --- | --- |
| Flat DQN | 62.65 | 329.03 |
| WS-Option | 101.21 | 12.56 |
| Director | 40.38 | 22.5 |
| Flat MuZero | 851.53 | 822.39 |
| **GoalZero** | 441.03 | 62.34 |

