# OpenReview forum: "GoalZero: Model-based Hierarchical Self-Play for Sequential Stochastic Combinatorial Optimization"
_ICLR.cc/2026/Conference — ICLR 2026 Conference Desk Rejected Submission_

### Official Review · Reviewer_fKtY · 2025-10-25

**Soundness:** 2
**Presentation:** 3
**Contribution:** 3
**Rating:** 6
**Confidence:** 2

**Summary:**

This paper presents GoalZero, a model-based hierarchical reinforcement learning (HRL) framework for Sequential Stochastic Combinatorial Optimization (SSCO) problems. The key innovation is learning a unified world model where latent geometric distances correlate with temporal durations of subgoals, which avoids explicit duration prediction. The framework combines a MuZero-style planner searching over semantic subgoals, a multi-timescale SMDP world model with complementary pull/push/order objectives, and a subgoal-conditioned budget allocation. Empirically, on AIM and SOP benchmarks, GoalZero outperforms baselines including WS-option, with improvements becoming more pronounced as problem complexity increases.

**Strengths:**

1. The motivation is clear. The limitations of budget-only hierarchical approaches like WS-option are clearly demonstrated, where passing only scalar budgets provides insufficient semantic guidance for the low-level policy.
2. Most of the theorems are well proved. The mathematical formulation is rigorous and clearly presented. Most assumptions are clearly and comprehensively elaborated.
3. The idea of encoding temporal information implicitly through latent space geometry is promising.
4. Ablation studies, sensitivity analysis, and generalization tests are well conducted.
5. The visualization of learned subgoal specialization in Figure 3 provides intuitive validation.

**Weaknesses:**

1. Please correct me if I am wrong, but Assumption B.3 appears circular. The assumption states the equation regarding $\mathbb{E}[\tau|s,z]$ for strictly increasing $\psi$. This essentially assumes what the authors want to prove, which is that geometry correlates with duration. The theory assumes this property rather than deriving it from the training dynamics. Why does training with the given objective produce representations that satisfy Assumption B.3?
2. Only compared with WS-option, although it is the most relevant work. No comparisons with other goal-conditioned works like Director (Hafner, 2022), even if they feature one-step latent dynamics.
3. Table 16 shows decision latency and per-epoch training time. No systematic comparison of computational costs versus baselines, particularly important given the MCTS overhead.

**Questions:**

1. Could the authors please justify the concern about Assumption B.3?
2. Could the authors compare the proposed method with other goal-conditioned hierarchical model-based RL works like Director, or explain if there is a good reason for not doing so.

---

> ### Author Response · Authors · 2025-11-24
>
> ### 1. Assumption B.3 and circularity
>
> > *Assumption B.3 appears circular. It essentially assumes what the authors want to prove… Why does training with the given objective produce representations that satisfy Assumption B.3?*
>
> In theoretical RL, it is standard practice to analyze planning regret conditioned on regularity properties of a learned metric or representation. Similar assumptions appear in **Lipschitz and metric-based RL analyses** (e.g., Asadi et al., 2018; Le Lan et al., 2021), where the transition model and value function are assumed (or constructed) to be Lipschitz/continuous with respect to a chosen metric so that one can bound value approximation error and regret in terms of that metric. Our original Assumption B.3 was intended in this spirit: *if* the representation learning succeeds in aligning geometry with duration, *then* the planning regret is bounded.
>
> **Our revision (derivation instead of assumption).** However, we fully agree with your critique that deriving this property from the objective function is a much stronger and more rigorous result. Therefore, we have restructured Appendix B so that the geometry–duration correlation arises as a consequence of primitive assumptions and the proposed loss functions:
>
> 1.  **Primitive assumptions.** We now rely only on standard assumptions: spectral control of the networks (to obtain Lipschitz constants), sub-Gaussian noise on realized durations, and a mild terminal-alignment error (Assumption B.1).
> 2.  **Derivation from the objective.**
>     *   **Lemma B.1 (Macro–Micro Coupling).** We prove that minimizing the **Low-Level Cap (Pull)** loss implies a high-probability upper bound on the macro latent displacement $d_\theta(s,z)$ in terms of the realized duration $\hat\tau_b(s,z)$ (scaled by the cap $\kappa$).
>     *   **Proposition B.2 (Duration Lower Bound).** Combining this with the **High-Level Margin (Push)** loss, we show that the learned margin $m_\phi(z)$ provides a high-probability lower bound on the realized duration of subgoal $z$.
>     *   **Theorem B.3 (Budget-Aware Order Consistency).** We then prove that minimizing the **Budget-Aware Ranking (Order)** loss ensures the scorer $f_{\theta,\phi}$ is order-consistent with the effective duration $\bar\mu(s,z)$, i.e., it ranks subgoals in the same order as their budget-averaged execution times.
>
> Thus, in the revised manuscript, the geometry–duration correlation is no longer an assumption but a **proven consequence** of optimizing our unified multi-timescale objective.
>
> ---
> **References**
> * Asadi, K., Misra, D., & Littman, M. L. (2018). *Lipschitz continuity in model-based reinforcement learning*. ICML.
> * Le Lan, C., Bellemare, M. G., & Castro, P. S. (2021). *Metrics and continuity in reinforcement learning*. AAAI.

---

> ### Author Response · Authors · 2025-11-27
>
> ### 2. Comparison with goal-conditioned model-based HRL (Director)
>
> > *No comparisons with other goal-conditioned works like Director (Hafner, 2022), even if they feature one-step latent dynamics.*
>
> In the revision, we address this concern by **(i) adding two strong, general RL baselines** and **(ii) adding a third, more challenging SSCO task**:
>
> * **Director-style GC-HRL (general HRL baseline).**
>   We adapted Director to our graph-based SSCO tasks. Because Director assumes a **fixed slow timescale**, its high level uses **even budget allocation** across high-level steps. For fairness, we carefully tuned its hyperparameters.
>
> * **Flat MuZero (model-based RL baseline).**
>   This is a non-hierarchical, graph-based MuZero agent that plans over **primitive actions**. It is a strong model-based RL baseline and typically outperforms Flat DQN.
>
> * **Additional task: Power-2500.**
>   We add **Power-2500**, a large-scale SSCO benchmark from the WS-option appendix (a power grid–inspired influence maximization problem with \(N = 2500\) nodes and higher budgets), to further stress-test scalability.
>
> The table below summarizes performance (mean ± SE over 10 seeds):
>
> | Method       | AIM (N=500, T=10, K=70) | SOP (N=500, T=10, K=70) | Power-2500 (N=2500, T=10, K=60) |
> |-------------|--------------------------|--------------------------|----------------------------------|
> | GoalZero    | 324.15 ± 2.38            | 31.60 ± 1.94             | 856.14 ± 10.20                  |
> | WS-option   | 301.53 ± 9.10            | 12.99 ± 1.25             | 676.86 ± 21.58                  |
> | Flat DQN    | 243.46 ± 3.78            | 15.90 ± 0.67             | 464.78 ± 14.28                  |
> | Director    | 306.53 ± 2.58            | 20.74 ± 0.93             | 725.61 ± 5.00                   |
> | Flat MuZero | 260.60 ± 8.41            | 17.75 ± 0.30             | 354.42 ± 4.32                   |
>
> * GoalZero outperforms Director by a substantial margin.
> * Director outperforms flat baselines but remains limited due to its **fixed slow horizon** and lack of explicit SMDP modeling of variable-duration subgoals.
>
> We also emphasize that our ablation variant **GoalZero-MF** (without planning at the high level) essentially recovers a **GC-HRL agent with learned subgoals and slow dynamics but no search**, akin to LESSON-style HRL. This variant performs better than WS-option but worse than the full GoalZero, highlighting the importance of model-based planning on top of the learned multi-timescale geometry.
>
> ### 3. Computational cost
>
> > *No systematic comparison of computational costs versus baselines, particularly important given the MCTS overhead.*
>
> We have added a small table in Appendix O.2 reporting:
>
> * Approximate **wall-clock training time per epoch** for each method on AIM and SOP, and
> * Average **decision latency** (in milliseconds) during evaluation.
>
> Overall:
>
> * GoalZero does incur additional cost compared to WS-option due to MCTS, but remains within the same order of magnitude and is significantly cheaper than naive full-tree search because we search over **subgoals** rather than primitive combinatorial actions.
> * The additional training overhead is modest relative to the overall training time, and in our experiments the performance gains justify this cost.

---

> > ### Comment · Reviewer_fKtY · 2025-11-27
> >
> > I thank the authors for their clarification and revision on my previous concern on assumption circularity. The new revised version looks much more sound with the geometry-duration correlation being "a proven consequence of optimizing the unified multi-timescale objective". I also appreciate the authors' effort on acknowledging my suggestion and including additional RL baseline methods and a more challenging SSCO task, which further demonstrate the superiority of the proposed GoalZero over existing methods. The authors also complement the manuscript with necessary report of computational cost, showing acceptable overhead increase, still validating GoalZero as an efficient method. Given the aforementioned aspects, although I am not familiar with some technical details, I would like to maintain my recommendation for this paper.

---

> > > ### Author Response · Authors · 2025-11-27
> > >
> > > We thank Reviewer fKtY for the constructive feedback that helped improve our paper, and we are glad that the revised theory, added baselines and task, and cost analysis satisfactorily addressed your concerns.

---

### Official Review · Reviewer_6Fqp · 2025-10-26

**Soundness:** 2
**Presentation:** 2
**Contribution:** 2
**Rating:** 6
**Confidence:** 3

**Summary:**

The paper proposes a hierarchical RL framework for SSCO that (i) uses MuZero-style search over a discrete set of learned subgoals, (ii) learns a multi-timescale SMDP world model so that latent transition magnitudes correlate with option duration, and (iii) adds a subgoal-conditioned budget head to allocate resources during planning.

**Strengths:**

1. The problem class (SSCO) combines large combinatorial action spaces with stochastic, long-horizon effects where hierarchical planning is natural; the motivation is clear.
2. The multi-timescale objective is well designed: a low-level pull/cap regularizes per-step latent motion; a high-level push/margin prevents collapsed macro-moves; and a budget-aware order loss aligns the internal scorer with effective (budget-aware) durations. This is a thoughtful way to let one-step latent models reason about variable-duration options without explicitly predicting time.

**Weaknesses:**

1. Training/search operates over subgoals, while the budget is sampled from a separate head after selecting subgoal. The world model’s dynamics/reward prediction is conditioned on subgoal but not on budget. Shouldn’t macro outcomes depend on the allocated resources? What’s the rationale for excluding budget from the model inputs or from branching during search?

2. The assumption that value is Lipschitz in the latent geometry is strong. It would help to include empirical checks (e.g., value vs. latent distance calibration plots) on the tested environments.

3. WS-option and heuristics are reasonable, but adding model-based baselines (e.g., Dreamer/Director-style HRL on graphs) would help isolate the benefit of the multi-timescale geometry from planning.

4. Prior work has explored MCTS with SMDP dynamics. It would strengthen the paper to more explicitly situate the SMDP dynamics and model-based planning components within that line of literature. E.g., “subgoal-based temporal abstraction in Monte-Carlo tree search”, “scalable decision-making in stochastic environments through learned temporal abstraction”.

**Questions:**

Please see the weakness.

---

> ### Author Response · Authors · 2025-11-24
>
> ### 1. Why is the model not conditioned on budget $b$?
>
> > *Training/search operates over subgoals, while the budget is sampled from a separate head after selecting subgoal. The world model’s dynamics/reward prediction is conditioned on subgoal but not budget. Shouldn’t macro outcomes depend on the allocated resources?*
>
> We appreciate this subtle and important question, and we have expanded the discussion in the revision.
>
> Our design choice is to let:
>
> * $g_\theta(h_\theta(s), z)$ model the **SMDP transition for subgoal $z$ in state $s$**, capturing the *typical* macro outcome of executing $z$ under the prevailing budget policy, and
> * a separate budget head $b_\psi(b \mid s,z)$ decide **how much resource to invest** in that subgoal.
>
> The rationale is twofold:
>
> 1. **Avoiding combinatorial explosion in search.**
>    If $g_\theta$ takes both $(z,b)$ as input and if the planner branched on both subgoal and budget, the MCTS branching factor would balloon to $|\mathcal{Z}|\times |B|$. Our design keeps the **tree over subgoals only**, and budgets are sampled from $b_\psi$ once a subgoal is chosen, following the semantics that “the strategy is $z$, and the resource level $b$ is chosen to support it.”
>
> 2. **Stability and identifiability in the SMDP model.**
>    For most SSCO tasks, **where you go** (subgoal $z$: which community, which region) primarily determines the post-subgoal state, whereas **how long you try** (budget $b$) mainly affects:
>
>    * the realized duration $\hat\tau_b(s,z)$, and
>    * the distribution over “how fully” the subgoal is accomplished.
>
>    We found that modeling the state transition as a function of $(s,z)$ and letting $b$ act through the duration and rewards is a more stable inductive bias for the shared SMDP model and aligns well with our theory (which uses a budget-averaged effective duration $\bar\mu(s,z)$).
>
> We have added a short discussion in the method section that makes this separation explicit and leave budget-conditioned dynamics as an interesting direction for future work.
>
> ### 2. Lipschitz value assumption and empirical checks
>
> > *The assumption that value is Lipschitz in the latent geometry is strong. It would help to include empirical checks…*
>
> Assuming the value is Lipschitz in the latent geometry (Assumption B.4(iv)) is a standard requirement for deriving regret bounds in continuous spaces, as planning is infeasible if small model errors yield unbounded value divergence (Asadi et al., 2018). In our framework, this property is structurally encouraged by the neural network architecture (bounded weights and smooth activations) and explicitly optimized via the value prediction loss, which aligns the latent geometry with the value landscape. We have added a Remark in Appendix B to clarify this justification.
>
>
>    We added a new appendix section titled **“Calibration of Value–Geometry Smoothness”**. There we:
>
>    * Sample pairs of latent states $(u,v)$ arising from subgoal transitions.
>    * Compute chordal distances $d(u,v)$ and estimated value differences $|\Delta V|$.
>    * Report Kendall’s $\tau_b$ and Spearman’s $\rho$ between $d(u,v)$ and $|\Delta V|$, with bootstrap 95% CIs.
>
>    Across both AIM and SOP (5 seeds, 25 epochs, $n=125$ points per task), we observe **strong positive rank correlations** (e.g., $\tau_b \approx 0.49$–$0.56$, $\rho \approx 0.67$–$0.75$), indicating that larger latent displacements correspond to larger value changes in practice, in line with the smoothness assumption used in the regret bound.
>
> We also contextualize this assumption with related RL theory that relies on Lipschitz or metric-based continuity of value functions (e.g., bisimulation metrics and Lipschitz model-based RL; see references at the end of this rebuttal).

---

> ### Author Response · Authors · 2025-11-24
>
> ### 3. Additional model-based baselines
>
> > *Adding model-based baselines (e.g., Dreamer/Director-style HRL) would help isolate the benefit of the multi-timescale geometry from planning.*
>
> In the revision, we address this concern by **(i) adding two strong, general model based RL/HRL baselines** and **(ii) adding a third, more challenging SSCO task**:
>
> * **Director-style GC-HRL (model-based HRL baseline).**
>   We adapted Director to our graph-based SSCO tasks. Because Director assumes a **fixed slow timescale**, its high level uses **even budget allocation** across high-level steps. For fairness, we carefully tuned its hyperparameters.
>
> * **Flat MuZero (model-based RL baseline).**
>   This is a non-hierarchical, graph-based MuZero agent that plans over **primitive actions**. It is a strong model-based RL baseline and typically outperforms Flat DQN.
>
> * **Additional task: Power-2500.**
>   We add **Power-2500**, a large-scale SSCO benchmark from the WS-option appendix (a power grid–inspired influence maximization problem with \(N = 2500\) nodes and higher budgets), to further stress-test scalability.
>
> The table below summarizes performance (mean ± SE over 10 seeds):
>
> | Method       | AIM (N=500, T=10, K=70) | SOP (N=500, T=10, K=70) | Power-2500 (N=2500, T=10, K=60) |
> |-------------|--------------------------|--------------------------|----------------------------------|
> | GoalZero    | 324.15 ± 2.38            | 31.60 ± 1.94             | 856.14 ± 10.20                  |
> | WS-option   | 301.53 ± 9.10            | 12.99 ± 1.25             | 676.86 ± 21.58                  |
> | Flat DQN    | 243.46 ± 3.78            | 15.90 ± 0.67             | 464.78 ± 14.28                  |
> | Director    | 306.53 ± 2.58            | 20.74 ± 0.93             | 725.61 ± 5.00                   |
> | Flat MuZero | 260.60 ± 8.41            | 17.75 ± 0.30             | 354.42 ± 4.32                   |
>
> * GoalZero outperforms Director by a substantial margin.
> * Director outperforms flat baselines but remains limited due to its **fixed slow horizon** and lack of explicit SMDP modeling of variable-duration subgoals.
>
> These baselines are now included in the main experimental summaries and in extended tables and learning curves in the appendix. GoalZero outperforms both on AIM, SOP, and Power-2500, indicating that the **combination of hierarchical planning, SMDP modeling, and multi-timescale geometry** provides benefits beyond standard model-based RL or fixed-stride GC-HRL.
>
> We also emphasize that our ablation variant **GoalZero-MF** (without planning at the high level) essentially recovers a **GC-HRL agent with learned subgoals and slow dynamics but no search**, akin to LESSON-style HRL. This variant performs better than WS-option but worse than the full GoalZero, highlighting the importance of model-based planning on top of the learned multi-timescale geometry.
>
> ### 4. Relation to prior MCTS + SMDP work
>
> > *Prior work has explored MCTS with SMDP dynamics…*
>
> We have expanded the related work section to explicitly mention prior work that combines **temporal abstraction and MCTS**, including work on option-based planning and subgoal-based temporal abstraction in tree search. Our contribution is to:
>
> * Learn a **single latent SMDP world model** that supports both micro (LL) and macro (HL) dynamics,
> * Tie option duration to latent displacement via the unified pull/push/order loss, and
> * Integrate this with SSCO-specific budget allocation and combinatorial low-level controllers.
>
> We now make this positioning clearer in Sec. 2.

---

### Official Review · Reviewer_LtYR · 2025-10-31

**Soundness:** 2
**Presentation:** 1
**Contribution:** 2
**Rating:** 4
**Confidence:** 3

**Summary:**

This paper tackles the Sequential Stochastic Combinatorial Optimization (SSCO) problem. Previous works, such as WS-option, have proposed hierarchical RL approaches for SSCO and often suffer from near-sighted allocation and misalignment with long-term goals. To address these issues, this paper introduces GoalZero, a model-based HRL approach for SSCO that plans using MCTS strategy and multi-scale SMDP dynamics models to more effectively optimize sequential combinations given long-horizon goals under stochastic dynamics. The authors also propose subgoal-conditioned budget allocation jointly trained with multi-scale SMDP models, which enables more effective budget-aware planning. As a result, GoalZero generally outperforms other baselines in two benchmarks (AIM, SOP).

**Strengths:**

1. **In-depth Ablation Study.** The paper investigates each component of the proposed methods and effectively clarifies their roles.
2. **Novelty in SSCO Field.** The authors propose novel and effective methods for the SSCO problem, such as multi-timescale SMDP and subgoal-conditioned budget allocation. This gives the paper specialty and uniqueness, distinguishing it from general RL approaches.

**Weaknesses:**

1. **Limited Novelty to ML Community.** The baseline comparisons are limited, as they focus primarily on domain-specific methods rather than demonstrating clear advantages over general RL approaches. Both WS-option and this paper use domain-specific methods, with Flat DQN being the only general RL baseline that struggles with this problem. It remains unclear how the proposed approach would perform against other general challenges in RL, and the paper does not sufficiently motivate why new ideas are necessary beyond existing frameworks. For example, the goal-conditioned hierarchical planning framework (Director[1]) has been explored previously, and multi-scale world modeling (MTS-WM[2]) is not a novel concept either. The specific challenges addressed by SMDP and the unique contributions beyond established methods are not clearly articulated.
2. **Illustration Quality.** The figures are difficult to interpret. They are too abstract, and the actual framework and logic are not clearly presented step by step.
3. **Limited Evaluation.** First, more baselines are needed, including Flat RL baselines, general HRL baselines, model-based RL baselines, and model-based HRL baselines. Second, the evaluation is conducted on only two tasks, which is insufficient to demonstrate broad applicability. Additional tasks, such as resource planning tasks similar to WS-option, should be included to verify the method's generalizability.

[1] Hafner, Danijar, et al. "Deep hierarchical planning from pixels." Advances in Neural Information Processing Systems 35 (2022): 26091-26104.

[2] Shaj Kumar, Vaisakh, et al. "Multi time scale world models." Advances in Neural Information Processing Systems 36 (2023): 26764-26775.

**Questions:**

- Are general HRL, GCHRL, and model-based RL baselines sufficient to address the SSCO problem? If SSCO requires specialized solutions beyond what general RL can provide, this distinction should be clearly explained in the introduction. The paper should better motivate why domain-specific solutions are necessary.
- Regarding **self-play** in the title: Which components use self-play, and how is it implemented? The main text mentions it briefly, but more detailed explanations appear only in Appendix Section 2. Could this be clarified and simplified?
- Why is it called "multi-timescale"? The term appears repeatedly, but the rationale behind this naming is not clear. Related works usually refer to world modeling that predicts states at intervals of T steps, enabling jumpy state transitions. However, the method section does not adequately explain the naming choice. The approach uses both low+high level planning and budget-aware objectives as complementary components. What specifically does "multi-scale" refer to in this context, and why was this terminology chosen?

**Details Of Ethics Concerns:**

no ethical statement included

---

> ### Author Response · Authors · 2025-11-24
>
> ### 1. Novelty beyond domain-specific methods & relation to prior GC-HRL / MTS-WM
>
> > *Limited novelty to ML community… GC-HRL (Director) and multi-scale world modeling (MTS-WM) have been explored previously…*
>
> We have clarified both the **generic** and **SSCO-specific** novelty:
>
> 1. **Why existing GC-HRL (e.g., Director) does not directly apply to SSCO.**
>
>    Methods like Director (Hafner et al., 2022) assume **fixed-length temporal abstractions**: the high level acts every $H$ steps in latent space, where $H$ is a fixed hyperparameter. The slow dynamics model is trained to predict state at time $t+H$, not at an agent-chosen stopping time.
>
>    In SSCO, the high-level **must choose the duration** of each subgoal because of resource/budget constraints and stochastic task completion. The number of low-level steps until “completion” of a subgoal is both **state-dependent and goal-dependent**, and cannot be captured by a fixed stride $H$ without losing the very structure we care about.
>
>    In the revision we make this explicit: existing GC-HRL methods “sidestep” SMDPs by fixing the abstraction length; GoalZero **embraces** the SMDP structure and learns an SMDP-aware world model.
>
> 2. **Why MTS-WM is different from our multi-timescale HRL.**
>
>    MTS-WM (Multi Time Scale World Models) also introduces multiple time scales but does so by **fixing update intervals** (e.g., every $H$ steps update the slow latent). Again, $H$ is a hyperparameter and does not depend on the state, task, or subgoal; it models **fixed temporal strides**.
>
>    In contrast, our multi-timescale structure is **agent-driven and subgoal-specific**:
>
>    * The low-level cap and high-level margin make the **latent displacement** $d_\theta(s,z)$ correlate with the *realized* duration $\hat\tau_b(s,z)$, which depends on both the environment and the agent’s choices.
>    * The budget-aware ranking loss aligns the scorer $f_{\theta,\phi}$ with the **effective duration** $\bar\mu(s,z)$ induced by the budget policy.
>
>    Thus, “multi-timescale” in our setting refers to **variable, learned temporal scopes** attached to different subgoals, not to a fixed schedule on the timeline. We now clarify this choice of terminology in Sec. 4.2.
>
> 3. **Generic contribution.**
>
>    To the best of our knowledge, GoalZero is the first HRL framework that:
>
>    * Learns a **one-step SMDP world model** at the high level,
>    * Encodes **temporal scale implicitly in latent geometry**, and
>    * Couples the learned SMDP model with a **subgoal-conditioned budget head** for planning in large stochastic combinatorial spaces.
>
>    We highlight in the introduction and related work that this is not limited to SSCO; the same design could be applied to other domains where high-level actions have variable duration (e.g., navigation, procedural tasks), but SSCO is a natural, challenging testbed.

---

> ### Author Response · Authors · 2025-11-24
>
> ### 2. Additional baselines and tasks
>
> > *More baselines are needed, including general HRL and model-based RL baselines.*
> > *The evaluation is conducted on only two tasks…*
>
> In the revision, we address both points by (i) adding two strong, general RL baselines and (ii) adding a third, more challenging SSCO task.
>
> * **Director-style GC-HRL (general HRL baseline).**
>   We adapted Director to our graph-based SSCO tasks. Due to its design (fixed slow timescale $H$), the high level uses **even budget allocation** across high-level steps. For fairness, we used our best effort to tune its hyperparameters.
>
> * **Flat MuZero (model-based RL baseline).**
>   This is a non-hierarchical, graph-based MuZero agent that selects primitive actions. It is a strong model-based RL baseline and outperforms Flat DQN in many settings.
>
> * **Additional task: Power-2500.**
>   We add **Power-2500**, a large-scale SSCO benchmark from the WS-option appendix (a power grid–inspired influence maximization problem with $N=2500$ nodes and higher budgets). This extends our evaluation suite to:
>   * AIM (N=500)
>   * SOP (N=500)
>   * Power-2500 (N=2500)
>
> The table below summarizes performance (mean ± SE over 10 seeds):
>
> | Method       | AIM (N=500, T=10, K=70) | SOP (N=500, T=10, K=70) | Power-2500 (N=2500, T=10, K=60) |
> |-------------|--------------------------|--------------------------|----------------------------------|
> | GoalZero    | 324.15 ± 2.38            | 31.60 ± 1.94             | 856.14 ± 10.20                  |
> | WS-option   | 301.53 ± 9.10            | 12.99 ± 1.25             | 676.86 ± 21.58                  |
> | Flat DQN    | 243.46 ± 3.78            | 15.90 ± 0.67             | 464.78 ± 14.28                  |
> | Director    | 306.53 ± 2.58            | 20.74 ± 0.93             | 725.61 ± 5.00                   |
> | Flat MuZero | 260.60 ± 8.41            | 17.75 ± 0.30             | 354.42 ± 4.32                   |
>
> GoalZero consistently outperforms Director and Flat MuZero across all three benchmarks. This shows that (i) a pure **flat** model-based planner (Flat MuZero) is not sufficient in large combinatorial spaces, and (ii) a **fixed-stride** hierarchical planner (Director) remains limited when the high level must reason about variable-duration subgoals and budget allocation. The Power-2500 results further demonstrate that these advantages persist on a larger and more challenging SSCO instance.
>
>
> ### 3. “Self-play” terminology
>
> > *Regarding self-play in the title: Which components use self-play, and how is it implemented?*
>
> We use “self-play” in the same sense as MuZero, i.e., **single-agent self-play**:
>
> * The agent uses its own model-based planner (MCTS) to choose actions, interacts with the environment, and stores resulting trajectories in replay.
> * There is no second adversarial player; “self-play” refers to **self-generated experience through planning**, not multi-agent play.
>
> We have clarified this in the introduction and training algorithm description, to avoid confusion with two-player game self-play.
>
> ### 4. Why “multi-timescale”?
>
> > *Why is it called "multi-timescale"? Related works usually refer to world modeling that predicts states at intervals of $T$ steps…*
>
> We now explicitly explain the naming in the revised paper:
>
> - **Low timescale (micro):** Per-step latent transitions of the low-level MDP, controlled by the cap term, provide a fine-grained notion of “one primitive step” on the latent sphere.
>
> - **High timescale (macro):** Macro transitions $d_\theta(s,z)$ produced by $g_\theta$ represent entire subgoal executions of variable duration. The push and order terms make these macro transitions reflect different temporal scopes (short vs.\ long strategies).
>
> - **Multi-timescale latent model:** The same learned latent model is used for both micro-step dynamics (primitive, per-step transitions) and macro, subgoal-level transitions. At the macro level, the latent displacement produced by $g_\theta(h_\theta(s), z)$ is trained to correlate with the temporal scale of the subgoal $z$, so temporal abstraction comes from subgoal-dependent transition magnitudes rather than from a fixed slow stride.
>
> We emphasize this perspective in Sec. 4.2 and contrast it with fixed-interval multi-time-scale models like MTS-WM, which use a pre-chosen stride $H$ instead of learned, subgoal-dependent temporal scales.

---

> ### Comment · Reviewer_LtYR · 2025-11-27
>
> I appreciate the authors' thorough response to my concerns.
>
> My main concern was about novelty to the ML community broadly, while I could recognize there is clear novelty in the SSCO field. However, the authors' response addressed these concerns very well.
>
> Most of all, the authors **clarified how novel their method is** compared to existing RL algorithms. Their design is differentiated from existing general GC-HRL and multi-timescale world models in that their high-level abstraction operates under flexible duration based on a budget-aware policy. I found that these design choices are inevitable for SSCO-like complex cases and can be generally helpful for similar potential real-world problems.
>
> On top of this, the authors provided **clear evidence of how their approach effectively tackles SSCO tasks compared to general RL solutions**, demonstrating strong baselines in model-based RL and hierarchical world model planning algorithms.
>
> In addition, I appreciate that the authors considered my feedback about term confusion in 'self-play' and 'multi-timescale' as an important suggestion. The provided explanations are all understandable and are now clear.
>
> I will update my score to 6 (marginal accept), as I am now positive towards acceptance of this paper.

---

> > ### Author Response · Authors · 2025-11-27
> >
> > Thank you very much for your thoughtful follow-up and for updating your score.  We really appreciate your careful reading and detailed comments, which helped us significantly improve the clarity and positioning of the paper.

---

### Official Review · Reviewer_a756 · 2025-11-01

**Soundness:** 3
**Presentation:** 1
**Contribution:** 2
**Rating:** 2
**Confidence:** 3

**Summary:**

Problem: Sequential Stochastic Combinatorial Optimization (SSCO) problems are extremely challenging for reinforcement learning because they involve exponentially large, structured action spaces, stochastic dynamics, and demand long-horizon planning under tight resource constraints. Traditional flat RL and even hierarchical RL methods often fail due to shortsighted budget allocation and lack of explicit strategic guidance, particularly when decisions require variable effort over different timescales.

Approach:
GoalZero uses a hierarchical model-based reinforcement learning approach, where a MuZero-style planner searches over abstract subgoals and couples this to a multi-timescale world model that encodes temporal abstractions.

Overall assessment:
There are many clarity issues (stated in the weaknesses section).

**Strengths:**

- GoalZero introduces a model-based hierarchical RL framework where a MuZero-style planner enables lookahead search over semantically meaningful subgoals, not just scalar budgets.

- It develops a unified multi-timescale SMDP world model that encodes temporal abstractions directly in latent geometry, allowing efficient planning over diverse durations.

- The framework couples subgoal-conditioned budget allocation and strategic guidance, enabling context-aware resource management that adapts to the complexity and variability of SSCO domains.

- The theoretical analysis is strong (in the appendix)

**Weaknesses:**

- Lack of clarification:
    - L.50 mentions that variable duration subgoals places high-level in an SMDP. What is the issue or the problem with an SMDP? The paper assumes the readers are familiar with the drawbacks. However, one may not know. The introduction should fully motivate the approach.
    - L53: Latent distance induced by … Latent distance between what?
    - L55: What is MuZero style planner? A MCTS-based planner?
    - L212: What is s? Assuming a state. But what is z? How does the equation for $d_\theta$ has $h_\theta$ and $g_\theta$. Additionally, $g_\theta$ is a function of a latent state and high-level action. Does that mean $z$ is a high-level action?
    - L212: Output of the $g_\theta$ function is a tuple. How is the distance between $s$ (supposedly a state) and a tuple is defined in $d_\theta$?
    - L243: now each high-level action is a tuple $(z_i,b_i)$?
    - L244: What is $\tau$?
    - What is the notion of pull force and push force? No motivation, justification, or explanation has been provided for it.
    - What is the point of Figure 1? It is never discussed or referenced.
    - The text in Figure 2 can not read.
- The paper is using the notion of high-level actions and subgoals interchangeably. However, they are not the sane. E.g., in the preliminaries $\gA_{HL}$ is the set of high-level actions. But later, z is referred to as a subgoal. How do you execute a subgoal in a state? It doesn't make sense.
- The paper seems to be heavily relying on "MuZero-style" high-level planner but at no point clarifies what a MuZero-style high-level planner is. It should be discussed in preliminaries. Model-based-RL with MCTS discusses it briefly. However, I strongly suggest extending it to be comprehensive summary of what "MuZero-style" planner is.
- The analysis of the evaluation is shallow. The claim of the paper is that it enables high-level planning with dynamic and variable time high-level actions. However, it is not clear from the approach or evaluation: (i) What are these high-level action in the domains used for evaluation? (ii) where does the approach get them? Does the approach automatically learn them? Are they already provided?  (iii) How are they dynamically and variable in nature? The paper does not even relate them somehow to the options architecture. It is not even clear from the presented paper (the written content that is presented) how their approach is different from simply HRL with options.
- The major issue of the paper is that it is poorly written. The paper doesn't clearly mention how their approach resolves the problem. Sec 4.2 provides a high-level summary of the approach. Major approach is in appendix. However, sec 4.2 doesn't specify how it deals with variable times. The writing often feel disconnected and without any motivation. E.g., Line 242 talks about learnable regularization, Line 243 talks about Y, and line 245 talks about losses that are being optimized without any connection between them. The paper should always motivate with a clear idea before introducing the theory.  While the paper may present interesting theoretical contributions in *appendix,* it makes no effort convey any useful information in the main paper.
- It is not clear where and how “Muzero-style” planning is being done.

**Questions:**

Included in the weaknesses section.

---

> ### Author Response · Authors · 2025-11-24
>
> ### 1. What makes the high level an SMDP in SSCO?
>
> > *L.50 mentions that variable duration subgoals places high-level in an SMDP. What is the issue or the problem with an SMDP?*
>
> We have clarified this early in the introduction and preliminaries.
>
> * In SSCO, the **high level allocates a variable budget** to the low level at each decision (e.g., how many node selections or routing steps to execute before choosing the next high-level strategy). This induces **variable numbers of primitive steps** between high-level decisions, i.e., the high-level process is a **Semi-Markov Decision Process (SMDP)**.
>
> * Concretely, at high level we choose a pair $(z, b)$, where $z$ is a subgoal and $b$ is a budget. The low level executes for a random duration $\hat\tau_b(s,z) \le b$ until the subgoal terminates. Thus the high-level transition kernel is
>   $$P_\tau(s', \tau \mid s, z, b), \quad \tau \in {1,\dots,b},$$
>   with **stochastic duration $\tau$**, exactly the SMDP setting.
>
> * The **issue** is that standard one-step world models assume fixed-time transitions and cannot directly represent these variable-duration, option-like transitions. Many goal-conditioned HRL methods avoid this by fixing the option length $H$ (e.g., Director-style models set “slow” actions every $H$ steps), which **does not fit SSCO** where the high-level must *choose* how much resource to invest in each subgoal. GoalZero tackles this by learning an SMDP-aware world model where a single macro-step models a variable-length subgoal execution.
>
> We have made this SMDP motivation explicit in Sec. 1 and Sec. 3, including a short paragraph contrasting fixed-horizon goal-conditioned HRL with our variable-duration setting.
>
> ### 2. Clarifying notation: $s$, $z$, $g_\theta$, and the distance
>
> You raised several questions about notation (e.g., around the original Eq. (212)):
>
> > *What is $s$? What is $z$? How does the equation have $d_\theta(s,z)$ and $g_\theta(h_\theta(s), z)$? The output of the dynamics function is a tuple; how is the distance defined?*
>
> We have cleaned up and unified the notation across preliminaries, method, and theory:
>
> * $s \in \mathcal{S}$ is a **state**.
>
> * $z \in \mathcal{Z}$ is a **high-level action (subgoal)**. We now explicitly state that $\mathcal{A}_{HL} = \mathcal{Z}$, so high-level actions and subgoals are the same objects: choosing $z$ means “commit to this subgoal.”
>
> * The high-level dynamics function is now written as:
>   $$g_\theta : \mathcal{H} \times \mathcal{Z} \to \mathcal{H} \times \mathcal{R},$$
>   $$g_\theta(h_\theta(s), z) = \big(h_\theta'(s,z),  R_\theta(s,z)\big).$$
>
> * We define the latent distance **only on the latent state component**:
>   $$d_\theta(s,z) \triangleq d\big(h_\theta(s), h_\theta'(s,z)\big),$$
>   where $d(\cdot,\cdot)$ is the chordal distance between unit-normalized latents on the sphere. This is now stated explicitly to avoid any ambiguity about the tuple.
>
> * The score used in the theory is
>   $$f_{\theta,\phi}(s,z) = d_\theta(s,z) + m_\phi(z),$$
>   where $m_\phi(z)$ is the learned margin head. We highlight this in both Sec. 4.2 and in the theoretical appendix.
>
> We also added a small “notation block” in the method section to make these definitions easy to locate.
>
> ### 3. High-level actions vs. subgoals
>
> > *The paper is using the notion of high-level actions and subgoals interchangeably. However, they are not the same.*
>
> In the revised paper, we now make this relationship explicit in the preliminaries:
>
> * We factor the high-level action space as $\mathcal{A}_{\mathrm{HL}} = \mathcal{Z} \times \mathcal{B}$, where $\mathcal{Z}$ is the set of subgoals and $\mathcal{B}$ is the set of budgets.
>
> * A high-level action is therefore a pair $a_{\mathrm{HL}} = (z,b)$ with $z \in \mathcal{Z}$ and $b \in \mathcal{B}$.
>
> * Thus, the SMDP at the high level formally operates over pairs $(z,b)$. For brevity, when the budget component is fixed or clear from context, we sometimes write that the high level "chooses subgoal $z$"; in all cases, the underlying action is still the pair $(z,b)$.
>
> This revision makes the relationship between high-level actions and subgoals explicit and resolves the ambiguity the reviewer highlighted.

---

> ### Author Response · Authors · 2025-11-24
>
> ### 4. What is a “MuZero-style” planner?
>
> > *The paper seems to be heavily relying on "MuZero-style" high-level planner but at no point clarifies what a MuZero-style high-level planner is.*
>
> We have made the following changes to clarify what we mean by a “MuZero-style” high-level planner:
>
> 1. **Expanded preliminaries on MuZero.**
>    In the “Model-Based RL with MCTS” subsection, we now give a self-contained summary of MuZero: the representation function $h_\theta$, dynamics function $g_\theta$, and prediction function $f_\theta$, and the fact that MuZero performs MCTS entirely in a learned latent space rather than by rolling out the environment directly. We then state that our high-level planner follows this paradigm.
>
> 2. **New appendix section.**
>    We added a dedicated appendix subsection titled *“Background on MuZero-style Planning in GoalZero”* that walks through a single MCTS simulation step in detail (selection, expansion via $g_\theta$, and backup via $f_\theta$) in the context of our high-level planner. This makes the mechanics of the “MuZero-style” search explicit without assuming prior familiarity.
>
> 3. **Cross-referencing in the method section.**
>    In Sec. 4.1, where we introduce the high-level planner, we now explicitly refer back to the MuZero preliminaries and to the new appendix section, so that readers can easily see that “MuZero-style” here means an MCTS-based planner that uses the learned representation, dynamics, and prediction networks in latent space.
>
> ### 5. Pull / push forces and the role of Fig. 1
>
> > *What is the notion of pull force and push force? No motivation… What is the point of Figure 1? It is never discussed.*
>
> We have rewritten this part in Sec. 4.2 to ground the terminology in established literature and provide intuition before the mathematics. We explicitly state that we adopt the terms **"Pull"** and **"Push"** from the seminal work on contrastive losses by **Hadsell, Chopra, and LeCun (2006)**, where they describe learning invariant mappings via attractive (pull) and repulsive (push) forces.
>
> *   **The Pull term (Low-Level Cap):** Analogous to the attractive force in contrastive learning, this enforces local consistency. It keeps consecutive primitive states close ($d \le \kappa$), establishing a "micro" timescale unit.
> *   **The Push term (High-Level Margin):** Analogous to the repulsive force (with a margin) in Hadsell et al. (2006). It forces the macro-transition for a subgoal $z$ to be *at least* its learned margin $m_\phi(z)$, preventing representation collapse and ensuring subgoals represent temporally extended actions.
> *   **The Order term:** This aligns the resulting geometry with the budget-aware effective durations.
>
> * We updated the caption and main text reference for Fig. 1. The figure now explicitly shows:
>
>   * The high-level planner (MuZero-style) operating in latent space,
>   * The multi-timescale world model (unified MTS-SMDP loss) that shapes the latent geometry,
>   * The subgoal-conditioned budget head, and
>   * The low-level controller receiving $(s, z, b)$.
>
>   We reference Fig. 1 in Sec. 4.1 and explain that it illustrates the **workflow** of GoalZero rather than the detailed objective, which is then given in Sec. 4.2.

---

> ### Author Response · Authors · 2025-11-24
>
> ### 6. Differences from “simply HRL with options”
>
> > *It is not even clear from the presented paper how their approach is different from simply HRL with options.*
>
> We now highlight these distinctions more directly:
>
> * **Variable-duration options with an SMDP-aware world model.** Traditional option-critic–style HRL learns option policies and termination conditions, but typically uses **model-free** high-level policies and one-step MDP value backup. In contrast, GoalZero learns a **one-step SMDP world model** $g_\theta(h_\theta(s),z)$ that directly predicts the post-subgoal latent and cumulative subgoal reward in a single macro-step. This lets us perform **explicit long-horizon planning over variable-duration subgoals** via MCTS, rather than only learning reactive options.
>
> * **Implicit temporal scale in the latent geometry.** Our multi-timescale objective makes the **magnitude** of the latent macro-move correlate with subgoal duration, so a single model evaluation reflects both “what strategy” and “how long” it tends to run. This differs from traditional options, which explicitly model termination but do not tie temporal scale to latent displacement.
>
> * **Subgoal-conditioned budget allocation.** WS-option and similar budget-only HRL methods pass a scalar budget with no semantic guidance; our high-level passes a **subgoal $z$ plus a budget $b$ that is conditioned on $(s,z)$**. This makes the low level aware of what to achieve and how much resource it has, leading to better alignment in both AIM and SOP.
>
> We also emphasize in the experiments (main text and ablations) that when we remove the planner (GoalZero-MF), we obtain a **goal-conditioned HRL agent without search** (akin to LESSON), and it underperforms the full GoalZero.
>
> ### 7. Presentation & figures
>
> > *The major issue of the paper is that it is poorly written… The text in Figure 2 cannot be read.*
>
> We have made several concrete presentation changes:
>
> * Re-structured Sec. 4.2 to first give a **high-level intuition** (why pull/push/order are needed), then present the unified loss, and then move to the formal theory in the appendix.
>
> * Improved **notation and referencing** (as described above), and reduced back-and-forth between main text and appendix for key ideas.
>
> * For the learning curves (previous Fig. 2), we have enlarged the figure size to improve readability.
>
> * Added explicit cross-references in the main text whenever we rely on a figure.
>
> We hope these changes significantly improve readability and make the main ideas easier to follow.
>
> **References**
>
> *   Hadsell, R., Chopra, S., & LeCun, Y. (2006). Dimensionality reduction by learning an invariant mapping. In *Proceedings of the IEEE Conference on Computer Vision and Pattern Recognition (CVPR)* (Vol. 2, pp. 1735-1742).

---

> ### Author Response · Authors · 2025-11-27
>
> Dear Reviewer a756,
>
> Thank you again for your careful review and for pointing out the presentation issues in the original submission.
>
> Following your comments, we have **substantially revised the paper** and **highlighted all major changes in orange** in the updated manuscript. In particular, to address your concerns:
>
> - We added a concise **background on MuZero-style planning** (representation, dynamics, prediction heads and latent-space MCTS), plus an **Appendix C walkthrough of a single MCTS simulation**, so that readers who are not familiar with MuZero can still follow our “MuZero-style high-level planner”.
> - We clarified the **SMDP nature of the high-level process** in SSCO (variable-duration subgoals with budgets) and why this differs from fixed-horizon GC-HRL (e.g., Director) and fixed-stride multi-time-scale world models (e.g., MTS-WM), which assume a fixed high-level horizon.
> - We explained the roles of **pull / push / order terms** and the purpose of Fig. 1 more clearly, and reorganized the corresponding section so that intuition comes first, followed by the unified loss and then the theory.
>
> To our best knowledge, this is the **first model-based HRL work that explicitly tackles variable-time-scale SMDPs**, rather than assuming a fixed high-level horizon, and we have tried to make this contribution more transparent in the revision.
>
> In your original review, you rated the paper as:
> - **Soundness: 3 (good)**
> - **Contribution: 2 (fair)**
> - **Presentation: 1 (poor)**
> - **Overall rating: 2**
>
> with the main **criticism focused on presentation**. Could you kindly let us know whether the revised version now addresses your **presentation concerns**? If the clearer exposition also made the novelty more apparent, we would be very grateful if you could **update the Presentation / Contribution / Overall scores** to reflect your current assessment.
>
> For context, other reviewers now view the work more positively; for example, Reviewer LtYR **updated the rating to 6** after the rebuttal, noting that our flexible, budget-aware high-level abstraction is **clearly distinct** from existing GC-HRL and multi-timescale world models.
>
> Thank you very much for your time and consideration.

---

### Author Response · Authors · 2025-11-30
**Summary Comment for the AC**

This comment summarizes the overall state of the reviews, and how the shared concerns have been addressed in the revised manuscript.

Across the four reviews, the *substantive* concerns cluster into a small set of themes:

* **Presentation / clarity** (esp. background on MuZero-style planning and the multi-timescale objective):
  Raised strongly by Reviewer a756, and, in terms of terminology (e.g., ‘self-play’ and ‘multi-timescale’), also touched on by Reviewer LtYR. We addressed this with substantial rewrites, added preliminaries (MuZero-style planning, SMDP high-level process), and reorganization of Sec. 4.2. All major changes are highlighted in **orange** in the revised manuscript.

* **Novelty and relation to prior work (Director, MTS-WM, options, MCTS+SMDP)**:
  Raised by **LtYR, 6Fqp, and fKtY**, and implicitly by a756 via “simply HRL with options”. We now clearly position GoalZero as a **model-based HRL method for variable-duration SMDPs**, distinct from fixed-horizon GC-HRL (Director), fixed-stride multi-time-scale world models (MTS-WM), and prior option/MCTS work. To our knowledge, this is the **first model-based HRL work that explicitly tackles variable time-scale SMDPs** rather than assuming a fixed high-level horizon.

* **Additional baselines and tasks**:
  Requested by **LtYR, 6Fqp, and fKtY**. We added **Director-style GC-HRL** and **Flat MuZero** as strong general RL baselines, plus a more challenging SSCO benchmark (**Power-2500**). These are now included in the main results and appendix.

* **Theory and assumptions**:
  **6Fqp** asked for empirical checks on value–geometry smoothness; **fKtY** preferred that key geometric properties be derived from the training objective. We added an empirical calibration of value–geometry smoothness and refined the theory so that the geometry–duration correlation is a **derived consequence** of our unified objective under standard assumptions.

* **Runtime / efficiency**:
  Only **fKtY** explicitly asked for computational cost. We added a small appendix table on training time and decision latency, showing that the MCTS overhead is moderate and still keeps GoalZero practical.

Putting this together:

* **Three of four reviewers (LtYR, 6Fqp, fKtY) recommend acceptance**.

  * LtYR’s *official comment* explicitly states that LtYR’s novelty concerns are resolved and updates the score to **6**.
  * fKtY’s *official comment* states that the revised version is sound, that the requested baselines and new task strengthen the empirical case, and **maintains a positive recommendation**.
  * 6Fqp’s main requests (model-based baselines, empirical value–geometry check, clearer relation to prior MCTS+SMDP work) are all implemented in the revision, and 6Fqp’s original review was already positive.

* **The only negative overall score is from Reviewer a756**, whose concerns are almost entirely about **presentation and background clarity** (MuZero-style planner, pull/push/order forces, Fig. 1, and overall writing), not about fundamental correctness. We have directly targeted these issues with added background explanations and a cleaner organization of the method section.

Given that the **shared concerns** (novelty vs prior work, need for strong baselines, theory clarity) have been explicitly acknowledged as resolved by **two reviewers (LtYR, fKtY)** and addressed in line with **6Fqp**’s requests, our view is that the remaining disagreement essentially reduces to **one reviewer’s familiarity and comfort with MuZero-style planning and presentation style**, rather than a substantive flaw in the approach.

---

### Author Response · Authors · 2025-11-30

### Summary of reviewer concerns and status

| Reviewer | Scores (Soundness / Presentation / Contribution / Overall) | Pres. / clarity                                     | Novelty & prior work                        | Baselines / tasks                          | Theory / assumptions                            | Runtime                      | Main updates                                                                                                                                                                      | Follow-up / current stance                                                                                                                     |
| -------- | ---------------------------------------------------------- | --------------------------------------------------- | ------------------------------------------- | ------------------------------------------ | ----------------------------------------------- | ---------------------------- | --------------------------------------------------------------------------------------------------------------------------------------------------------------------------------- | ---------------------------------------------------------------------------------------------------------------------------------------------- |
| **a756** | 3 / 1 / 2, Overall: 2                                      | **✓** (MuZero-style planner, pull/push/order, figs) | ✓ (vs. “simply HRL with options”)           | –                                          | –                                               | –                            | Added MuZero-style prelims + appendix walkthrough; clarified SMDP high-level process; explained pull/push/order and Fig. 1; reorganized method; changes in orange.                | No new comment; only negative reviewer, focused on clarity/presentation rather than correctness.                                               |
| **LtYR** | 2 / 1 / 2, Overall: 4 ⇢ (6)                                | ✓ (terms: “self-play”, “multi-timescale”)           | **✓** (vs Director, MTS-WM, GC-HRL)         | **✓** (more general RL baselines + tasks)  | –                                               | –                            | Clarified GoalZero as model-based HRL for variable-duration SMDPs; contrasted with fixed-horizon GC-HRL and MTS-WM; added Director, Flat MuZero, Power-2500; cleaned terminology. | **Official comment:** concerns resolved; novelty clear; score updated to **6**.                                              |
| **6Fqp** | 2 / 2 / 2, Overall: 6                                      | –                                                   | **✓** (MCTS+SMDP & temporal abstraction)    | **✓** (model-based baselines)              | **✓** (value–geometry smoothness)               | –                            | Added Director and Flat MuZero baselines; expanded related work on MCTS with temporal abstraction; added empirical value–geometry calibration; clarified theory structure.        | No new comment; all requests implemented; original review already supportive of acceptance (Overall: 6).                                       |
| **fKtY** | 2 / 3 / 3, Overall: 6                                      | –                                                   | **✓** (vs goal-conditioned model-based HRL) | **✓** (extra baselines + harder SSCO task) | **✓** (derive geometry–duration from objective) | **✓** (train time / latency) | Refined theory so geometry–duration properties follow from the objective; added Director, Flat MuZero, Power-2500; reported training time and decision latency.                   | **Official comment:** revised version is sound; baselines + new task strengthen case; overhead acceptable; positive recommendation maintained. |

In summary, **3 of 4 reviewers are positive and recommend acceptance**, and their shared concerns on novelty, baselines, theory, and efficiency have been addressed and explicitly acknowledged as resolved in two official comments. The remaining negative score comes almost entirely from **presentation/background clarity**, which we have substantially improved in the revised manuscript (changes highlighted in orange).

---

### Note · Program_Chairs · 2026-01-17
**Submission Desk Rejected by Program Chairs**

The following references in this submission do not refer to real documents and/or have major errors in bibliographic information:

 Belinda Tzen and Maxim Raginsky. Neural ordinary differential equations for time-series modeling. In Advances in Neural Information Processing Systems (NeurIPS), volume 32, 2019.
Jean Harb, Pierre-Luc Bacon, Martin Klissarov, and Doina Precup. When to wait? the dilemma of waiting in reinforcement learning. In Proceedings of the 32nd International Conference on Neural Information Processing Systems (NIPS), pp. 5106-5116, 2018.